# TIME-TO-EVENT PRETRAINING FOR 3D MEDICAL IMAGING

**Zepeng Huo**[1,*] **Jason Alan Fries**[1,*] **Alejandro Lozano**[2,*]
**Jeya Maria Jose Valanarasu**[3,5], **Ethan Steinberg**[1,6], **Louis Blankemeier**[2],
**Akshay S. Chaudhari**[2,5,9,10], **Curtis Langlotz**[4,5,8,10], **Nigam H. Shah**[4,5,7,8,9,11]

[1]Center for Biomedical Informatics Research, Stanford University
[2]Department of Biomedical Data Science, Stanford University
[3]Department of Computer Science, Stanford University
[4]Stanford Medical Center, Stanford Health Care
[5]Stanford Center for Artificial Intelligence in Medicine and Imaging
[6]Prealize Health
[7]Technology and Digital Solutions, Stanford Health Care
[8]Department of Medicine, Stanford School of Medicine
[9]Human-Centered Artificial Intelligence Institute, Stanford University
[10]Department of Radiology, Stanford University
[11]Clinical Excellence Research Center, Stanford School of Medicine
`{zphuo, jfries, lozanoe}@stanford.edu`

## ABSTRACT

With the rise of medical foundation models and the growing availability of imaging data, scalable pretraining techniques offer a promising way to identify imaging biomarkers predictive of future disease risk. While current self-supervised methods for 3D medical imaging models capture local structural features like organ morphology, they fail to link pixel biomarkers with long-term health outcomes due to a *missing context problem*. Current approaches lack the temporal context necessary to identify biomarkers correlated with disease progression, as they rely on supervision derived only from images and concurrent text descriptions. To address this, we introduce *time-to-event pretraining*, a pretraining framework for 3D medical imaging models that leverages large-scale temporal supervision from paired, longitudinal electronic health records (EHRs). Using a dataset of 18,945 CT scans (4.2 million 2D images) and time-to-event distributions across thousands of EHR-derived tasks, our method improves outcome prediction, achieving an average AUROC increase of 23.7% and a 29.4% gain in Harrell's C-index across 8 benchmark tasks. Importantly, these gains are achieved without sacrificing diagnostic classification performance. This study lays the foundation for integrating longitudinal EHR and 3D imaging data to advance clinical risk prediction.

## 1 INTRODUCTION

Foundation models for medical imaging have the potential to transform healthcare by assisting doctors in complex clinical decision making (Saab et al., 2024; Sox et al., 2024) and identifying novel pixel biomarkers predictive of future disease risk (Pai et al., 2024; Sriram et al., 2021). Such models rely on self-supervised learning (SSL) for obtaining supervision at the scale required to train on growing collections of 3D data (e.g., CT scans, MRIs). SSL captures local structural features by leveraging pretraining signal directly from images or cross-modal pairs (e.g., images and their text descriptions) (Zhang et al., 2022). While excelling at segmentation tasks and diagnostic classification of pathologies (Pinto-Coelho, 2023), SSL fails to learn prognostic biomarkers because the current pretraining regimens suffer from a *missing context problem* (see Figure 1). This issue arises when supervision sources are restricted to narrow time windows around the image, thus excluding

---

*Authors contributed equally. A detailed contribution breakdown is in Appendix C.

long-term temporal patterns that are correlated with disease progression, which limits a model's ability to identify prognostic biomarkers.

These long-term temporal patterns, which we refer to as longitudinal context, are readily available in a patient's electronic health record (EHR) which is routinely used by clinicians to guide the interpretation of images and inform treatment planning (Leslie et al., 2000; Holste et al., 2024). Longitudinal EHRs contain temporal information about the progression of disease, as well as years of patients' health outcomes. However, the full breadth of these outcome data, in terms of temporal structure and task diversity, is rarely used as a source of supervision when training image foundation models. Current approaches for training 3D imaging models typically restrict labels to diagnosis codes sourced from the same or nearby temporal context as the image and its textual description (Blankemeier et al., 2024). Because EHR data is readily available, it offers an untapped resource for large-scale pretraining of medical image models in a manner that uses long-term temporal context. More generally, image foundation models must reflect the settings in which they will be used, which is to assist in prognosis in the clinic (Negro-Calduch et al., 2021; Yala & Hughes, 2023).

However, performing effective risk estimation using imaging models involves navigating several challenges. Developing such models requires capturing correlations between pixels and outcomes spanning years, which is difficult with current SSL methods. Direct approaches to identifying pixel-level biomarkers, such as sequential image capturing (Lu et al., 2021; Bera et al., 2022), are difficult to scale for collecting large, high-quality datasets. Moreover, conducting risk estimation requires addressing right censoring, where the outcome of interest remains unobserved by the study's end. Naively excluding censored patients introduces bias and reduces available training data.

In medicine, time-to-event (TTE) modeling (also known as survival modeling) is commonly used to estimate future risk of an outcome at a specific time point conditioned on feature representations. Although TTE models offer many theoretical advantages, including the ability to estimate instantaneous risk at any given time point (Collett, 2023) and naturally handling right censoring (Kleinbaum & Klein, 1996), their use in image pretraining remains underexplored. Prior deep learning studies exploring TTE modeling in medical imaging have been restricted to small-scale, single-task applications, typically using 2D, end-to-end models (Zhu et al., 2016; Shu et al., 2021; Lu et al., 2021). 3D medical imaging data remains an underutilized resource, offering a wealth of biomarkers that could enhance clinical decision tools, particularly for the opportunistic detection of underdiagnosed conditions (Aali et al., 2024). However, large-scale TTE pretraining for 3D imaging has not yet been investigated, likely because multimodal medical datasets linking 3D images with longitudinal EHR data have only recently become available (Huang et al., 2024).

In this work, we propose *time-to-event pretraining* for medical imaging models as a way to address the missing context problem. Our central claim is that temporal supervision, defined by TTE distributions sourced from longitudinal EHRs, provides a readily available, scalable source of contextual information for pretraining that better captures prognostic pixel biomarkers. Moreover, by naturally handling right-censorship, TTE-based methods improve data efficiency and mitigate censorship bias. Our contributions are as follows:

- We present the first large-scale evaluation of time-to-event pretraining for 3D medical imaging encoders. We use a public dataset of 18,945 chest CT scans (equivalent to 4.2 million 2D images) linked to longitudinal EHR data containing 225M clinical events with a median follow-up time of 5 years.

- Our approach converts longitudinal EHR data into a source of time-to-event supervision, thus predicting not only if a clinical event will occur but also when. This richer pretraining signal goes beyond diagnostic classification explored in prior work and enables generating many pretraining tasks (8,192 in this work) that capture the temporal event structure available in longitudinal EHR data. This choice also increase per-image label density by an average of 3 times over prior approaches.

- Our approach substantially improves performance in predicting future medical outcomes, achieving on average a 23.7% increase in AUROC and a 29.4% improvement in Harrell's C-index over baseline models for 8 benchmark tasks without negatively impacting diagnostic classification performance in 8 external tasks. Our approach also improves model calibration, measured by the Integrated Brier Score, by an average of 54%.

All our experiments are conducted using public medical datasets to ensure full reproducibility. We also make all of our experiment code and pretrained model checkpoints available for download [1] [2], to contribute to the community for continued pretraining of imaging foundation models with prognosis as added benefit.

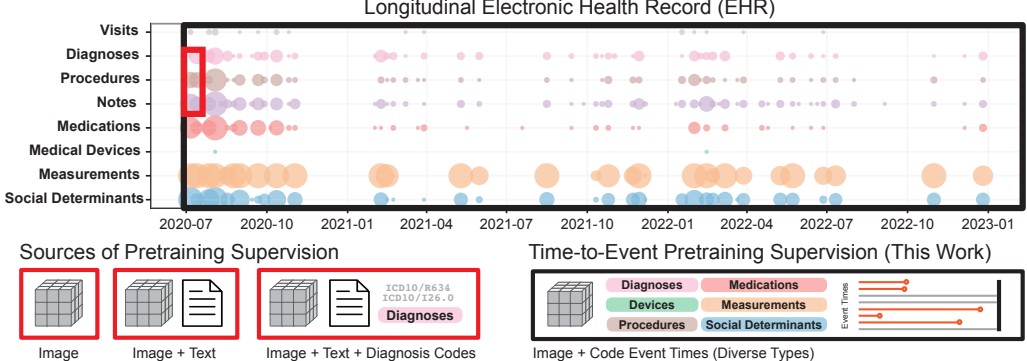

Figure 1: The *missing context problem* in medical imaging. Existing supervision sources (**red boxes**) are localized to the image itself (i.e., pixel features and descriptions of those features via text) or immediate clinical context via diagnosis codes. Doing so misses future information on disease progression (**black boxes**), which reduces the ability to learn correlations necessary for identifying prognostic pixel biomarkers. *Time-to-event pretraining* provides a principled framework for incorporating the vast amount of temporal supervision available in EHR data to estimate future risk in the presence of right censorship as well as leverage a large, diverse number of clinical tasks, beyond just diagnoses, for pre-training.

## 2 RELATED WORK

**Time-to-Event Modeling with Medical Images**  Time-to-event (TTE) modeling, also known as survival analysis, predicts the distribution of time until a specific event occurs, such as death. TTE models primarily include accelerated failure time models, which assume various probability distributions, e.g., exponential (Saikia & Barman, 2017), Weibull (Breheny, 2015), and Cox proportional hazard (Cox-PH) models (Cox, 1972) which is a semi-parametric approach with a constant hazard ratio assumption. Non-parametric models like random survival forests (Ishwaran et al., 2008) capture non-linear interactions. In the deep learning era, methods such as DeepSurv (Katzman et al., 2018) and MOTOR (Steinberg et al., 2024) provide higher level feature learning, making TTE modeling easier to extend to complex inputs such as medical images. Prior TTE methods for imaging assume 2D and 2.5D model architectures and have focused on end-to-end training for small-scale, single-task models. DeepConvSurv (Zhu et al., 2016) was the first work to replace the log-partial hazard (the exponential component of a Cox model) with a CNN, enabling survival prediction directly from 2D images. Similarly, Shu et al. (2021) and Lu et al. (2021) modeled the log-partial hazard with a CNN-RNN to encode sequential images.

**Pretraining for 3D Medical Image Models**  Although early work in medical imaging relied on supervised pretraining using general-domain datasets such as ImageNet (Xie & Richmond, 2018; Ke et al., 2021), self-supervised learning (SSL) is now the predominate approach to scaling pretraining in medical imaging models. Popular approaches include reconstruction or de-noising objectives via masked autoencoders (MAE) (He et al., 2022), contrastive losses defined over paired samples (Chen et al., 2020; Sowrirajan et al., 2021) or leveraging multimodal pairs, such as medical images and their aligned text descriptions (Radford et al., 2021; Zhang et al., 2022) or unpaired medical images and text (Wang et al., 2022). SSL methods face added challenges when extended to 3D imaging, where instances (e.g., CT scans) contain over 100 times more pixel data than 2D counterparts like radiographs. Tang et al. (2022) combined unimodal contrastive learning, masked volume in-painting, and

---

[1] https://github.com/som-shahlab/tte-pretraining
[2] https://huggingface.co/StanfordShahLab

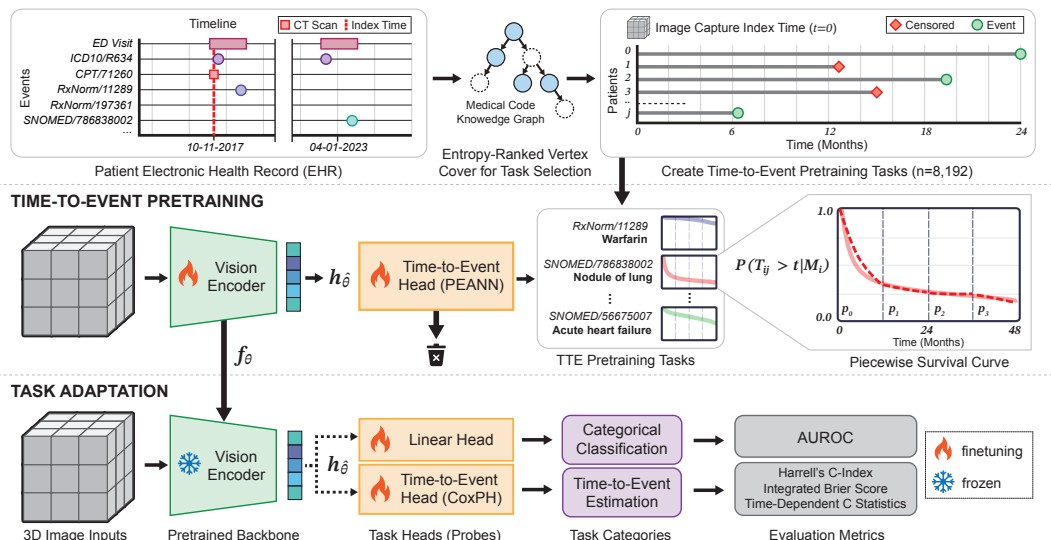

Figure 2: Overview of the proposed *time-to-event pretraining* pipeline. Patients' longitudinal EHR timelines are transformed into large-scale, time-to-event (TTE) pretraining tasks. These tasks, which reflect informative temporal patterns for medical outcome prediction, are then used for continued pretraining (full fine-tuning) of a 3D vision encoder. The resulting encoder is then frozen and adapted to downstream tasks via different task heads for classification or TTE estimation.

image rotation prediction to learn 3D structural information. Chen et al. (2023) explored 3D masked image modeling, demonstrating faster convergence compared to simple contrastive methods (SimCLR and MAE). Valanarasu et al. (2023) used a reconstruction loss to restore 3D CT volumes from tokens corrupted by noise, downsampling, and local masking. Finally, Blankemeier et al. (2024) introduced Merlin, a dual-objective framework that leverages EHR data by first using contrastive learning to align radiology reports with CT volumes, followed by training on disease phenotype classification using labels derived from diagnostic codes recorded during contemporaneous hospital visits.

Current SSL approaches excel at capturing structural features in medical images, such as organ morphology for image segmentation. However, these learned features are static and largely fail to identify dynamic patterns and biomarkers that are predictive of future health risks. Our contributions help bridge this gap in several key ways by leveraging future medical events to guide pretraining, resulting in a more powerful image encoder for outcome prediction tasks. First, we employ a TTE objective to capture future patient health dynamics while accounting for right censoring, leveraging longitudinal EHR data to greatly expand the scale, diversity, and temporal scope of pretraining supervision. Second, we evaluate pretraining using native 3D imaging architectures which are underexplored in prior TTE work. Finally, we evaluate the impact of TTE modeling in imaging at a larger scale than prior work, utilizing 18,945 CT scans—equivalent to 4.2 million 2D images—and 8,192 unique TTE tasks derived from longitudinal EHRs.

## 3 PRELIMINARIES

**Time-to-Event Modeling.** The objective of time-to-event modeling is to estimate the distribution of times $T$ until an event of interest occurs. Observable data is denoted as $\mathcal{O} = \left\{ (\tilde{T}_i, \Delta_i, X_i^T) : i = 1, ..., n \right\}$, where $X_i$ are the features of observation $i$ and $\Delta_i = \mathbb{I}(T_i \leq C_i)$ is an event indicator function whose value is 1 when the actual event time is observed. One complexity is that medical data is often right-censored, where the survival time $T$ is not observed due to a loss of followup. With right-censoring, we do not observe $T$ and instead observe time $\tilde{T} = \min(T, C)$, where the $C$ is the censoring time. When $\Delta_i = 0$, we do not know the true survival time, but we know that it

is greater than $\tilde{T}_i$. The mixture of known survival times and censored survival times necessitates methods that can estimate $T$ in an unbiased manner despite the censorship.

Different time-to-event models use different definitions to model the instantaneous hazard $\lambda(t)$, where $t$ represents the continuous time at which the hazard rate is evaluated. We use a piecewise exponential function (Kitchin et al., 1983) that splits the timeline into $P$ distinct intervals, or *pieces*, each with a constant hazard rate $\lambda(t)$. Each piece $p(t)$ has $\lambda_{ip(t)} = C_{ip}$, where $C$ depends on the patient's CT image $i$ and the specific interval $p$.

**Piecewise Exponential Neural Network.** To make use of deep learning models, we use a neural network to define $C_{ip}$. Using a piecewise exponential neural network (PEANN) (Fornili et al., 2014) greatly simplifies large-scale pretraining compared to Cox PH-based methods, which require creating batches where samples are paired with at least one uncensored patient (Harrell Jr et al., 1996). We first use a neural network to derive a CT image representation $M_{ip}$ for image $i$'s information at time piece $p$. We then apply a linear layer followed by an exponential function to define hazard $\lambda_{ip} = C_{ip} = \exp(A * M_{ip} + b)$ where $A$ and $b$ are learned parameters. With the hazard calculated, we can then input it into the survival function:

$$S_i(t) = \prod_{p=1}^{P} \exp\left[-\lambda_{ip}(\min(t, E_p) - S_p)\mathbb{I}(t \geq S_p)\right] \tag{1}$$

where each time piece $p$ has a starting point $S_p$ and end point $E_p$ and $\mathbb{I}(\cdot)$ is an indicator function. The standard survival likelihood loss function for image $i$ is then:

$$\mathcal{L}_i = [S_i(t)]^{1-\Delta_i}[f_i(t)]^{\Delta_i} \tag{2}$$

where $\Delta_i$ is event indicator whose value is 1 when the actual future event is observed for image $i$ and $f_i$ is the probability density function $f_i = -\frac{\partial S_i}{\partial t}$.

A full derivation of the loss function, including calculating the derivative and plugging in the survival function, can be found in Appendix M.

## 4 TIME-TO-EVENT PRETRAINING

In this work, we are interested in training a 3D image encoder to learn representations optimized for estimating the distribution of event times for future clinical outcomes. Using years of follow-up EHR data after image capture, we generate time-to-event pretraining tasks that provide large-scale training signal for estimating this distribution. We outline our approach for time-to-event pretraining in Figure 2. Given the computational costs of pretraining 3D image architectures from scratch, we evaluate the benefits of TTE supervision via continued pretraining of an existing neural network $f_\theta$, where $\theta$ represents the parameters of the pretrained backbone.

**Creating TTE Pretraining Tasks** EHR data captures a vast amount of structured information on patient demographics, diagnoses, procedures, medications, medical devices, social determinants of health, and other aspects of medical care. These data are encoded as timestamped, standardized identifiers called *medical codes* that map to ontologies (e.g., ICD10, RxNorm, CPT). These ontologies collectively represent a knowledge graph in the form of a directed acyclic graph (DAG). By treating each code as a separate task and organizing EHR data chronologically, we generate extensive training signals to model longitudinal health trajectories. This approach enables tasks such as predicting when a patient might develop lung cancer or be prescribed warfarin (a blood thinner used to prevent and treat blood clots), represented as time-to-event distributions conditioned on image features. However, naively treating each medical code as a task leads to millions of candidate pretraining tasks (4.3 million in our dataset). Many of these tasks are low-frequency or otherwise redundant given an ontology structure. To select pretraining tasks, we follow Steinberg et al. (2024)'s conditional Shannon entropy selection measure, which treats ontology-aware task selection as a vertex cover problem (West et al., 2001). We select a set of medical codes that maximizes conditional entropy given a task budget, ontology DAG, and frequency distribution of observed medical codes.

We then define our TTE pretraining procedure by predicting the time until the first occurrence (if there are multiple recurring ones) of a medical code as defined above, as shown in Appendix B.

The survival function in each time piece is modeled from the time piece's starting time, where the first time piece's starting time is the index time (here the time of the CT scan exam). When there is no event in a time piece the loss will be 0. Each TTE task (n=8,192) is modeled independently (Appendix O). We also apply censorship at patient death, which is the only competing risk. Given TTE labels for input image $X_i$, we use the loss described in Eq. 2 for full fine-tuning of $f_\theta$.

**Task Adaptation.** Once TTE pretraining is complete, we use the frozen encoder $f_\theta$ to generate feature embeddings for each input image $X_i$, such that $Z_i = f_\theta(X_i)$. These embeddings are then passed to a task-specific head $h_{\hat{\theta}}$, which can either be a classification head or a time-to-event head, depending on the task. We found that a CoxPH task head (DeepSurv), which directly optimizes for Harrell's C-index, performed better in practice than fine-tuning a PEANN task head, thus we used CoxPH for all TTE task adaption. The classification head outputs prediction probabilities $p_i(\hat{y}|Z_i)$ for discriminative tasks, while the survival head produces a time-dependent hazard score $H_i(t|Z_i)$ for time-to-event (TTE) tasks. The model's outputs are evaluated using task-appropriate metrics, e.g., AUROC for classification or Harrell's C-index and time-dependent C-statistics for TTE tasks.

## 5 EXPERIMENTS

**Hypotheses.** Our experiments measure the impact of TTE pretraining on an image encoder's ability to generate representations useful for medical prognosis. We explore the following hypotheses:

1. TTE supervision improves data efficiency by utilizing temporal information from future EHRs and censored patients, increasing available task labels per-pretraining instance.

2. Existing supervised and SSL pretraining methods struggle to learn strong pixel biomarkers for disease risk due to missing temporal connections with pathologies that will be detected in the future. TTE supervision, in contrast, provides a simple approach to leveraging complex temporal information found in longitudinal EHRs for purposes of learning prognostic pixel biomarkers.

3. TTE supervision, when conducted as post hoc, continued pretraining, does not negatively impact performance on standard categorical classification tasks as used for diagnostic image labeling.

**Setup.** All image and EHR preprocessing details are outlined in Appendix A. For our PEANN, we use 8 piecewise time bins and 8,192 pretraining tasks (see Appendices O,P) per the best-performing hyperparameters from Steinberg et al. (2024). Each time bin is created by uniformly dividing the range between the cohort's earliest and latest timestamps. We utilized two compute nodes from an on-premise cluster, each with 24 Intel Xeon 2.7GHz CPU cores, 200 GB RAM and 4 Nvidia A100 GPUs or 4 H100 GPUs.

|  | INSPECT | RSPECT |
|---:|:---:|:---:|
| # Patients | 19,402 | 7,279 |
| # Train | 18,945 | 5,823 |
| # Valid | 1,089 | 364 |
| # Test | 3,214 | 1,092 |
| Imaging | ✓ | ✓ |
| EHR | ✓ | ✗ |
| TTE Tasks | ✓ | ✗ |
| Diag. Tasks | ✓ | ✓ |
| Scan Type | Chest CT | Chest CT |

Table 1: Dataset summary statistics.

**Datasets & Evaluation Tasks.** We use two 3D medical imaging datasets (see Table 1). INSPECT (Huang et al., 2024) is a multimodal dataset of paired CT scans and radiology notes where each patient is linked to their longitudinal EHR [3] [4]. This provides an average of 5 years follow-up data post-CT scan and, in aggregate, contains 225 million medical events. RSPECT (Colak et al., 2021) is an image-only dataset of CT scans annotated by radiologists for imaging biomarkers related to pulmonary embolism and cardiac function. We evaluate 3 task categories in this work:

- *Prognostic TTE*: Estimate the distribution of event times for a specific outcome (e.g, mortality). We predict the first occurrence of an event, as the first occurrence (or only occurrence in cases such as mortality) is of higher clinical utility than subsequent events.

---

[3] https://stanfordaimi.azurewebsites.net/datasets?term=INSPECT
[4] https://redivis.com/datasets/dzc6-9jyt6gapt

- *Prognostic Classification*: Binarized classification formulation of TTE tasks using bucketed time bins. We use 1, 6, and 12 month bins. Unlike in TTE, censored patients are excluded in this category.
- *Diagnostic Classification*: Standard classification using diagnostic image label categories.

INSPECT defines 3 prognostic binary tasks: hospital mortality, hospital readmission, and pulmonary hypertension (PH); and 1 binary diagnostic classification task (pulmonary embolism). Using the provided EHR data we define 5 additional prognostic TTE tasks for lung pathologies: ATX (Atelectasis), CMG (Cardiomegaly), CONS (Consolidation) EDM (Edema), and PEFF (Pleural Effusion). These tasks were selected based on their use in common chest medical imaging datasets (Irvin et al., 2019). Appendix B details how labels were assigned. RSPECT provides whole-volume labels for 9 diagnostic tasks, but no longitudinal outcome data. RSPECT does not provide a public test set, so we impose a 80/5/15 split on train (n=7,279) for our experiments.

**Architectures.** We evaluate three model architectures: SwinUNETR, DenseNet, and ResNet. SwinUNETR (Tang et al., 2022) was originally designed for medical image segmentation tasks, combining elements from the Swin Transformer and the UNETR (U-Net Transformer) architectures. Weights are learned using a reconstruction loss on 10,050 3D brain/chest CT and MRI data (Valanarasu et al., 2023), a much larger pretraining dataset than used by other public 3D models, e.g., Wasserthal et al. (2023). Following Merlin, we adapt DenseNet-121 (Huang et al., 2016) and ResNet-152 (He et al., 2015) by inflating their 2D pretrained ImageNet weights (specifically the filters and pooling kernels) as described in Carreira & Zisserman (2017). Note that ResNet and DenseNet parameters were initialized using 2D weight inflation, but the architecture is fully 3D, using 3D convolutions in each Res-block or Dense-block, different from 2.5D methods (Hung et al., 2024). This process enables us to input 3D CT images into the models for training. We also evaluated Merlin's pretrained ResNet-152 backbone, but found it performed similar to our base ResNet-152 (see Appendix E), thus we use the base weight-inflated model for consistency across experiments.

**Model Baselines.** We evaluate the following continued pretraining approaches on all architectures:

- `base`: Baseline performance of the 3D pretrained SwinUNETR, DenseNet-121, and ResNet-152 models without continued pretraining on INSPECT. See Appendix H for a summary of the source pretraining datasets.
- `base/MTL`: Continued pretraining of the base model via multitask, supervised learning using the 8 INSPECT evaluation task labels. This controls for exposure to our training dataset in a consistent manner across architectures.
- `base/visit`: Continued pretraining using the same 8,192 tasks used for TTE supervision, but restricting label assignment to the same visit as the CT scan. This ablates the TTE component to learn temporal information and aligns more closely with the EHR-based supervision used by Merlin.
- `base/TTE`: Continued pretraining using TTE labels for 8,192 tasks occurring after the CT scan index timestamp.

All models except `base` use the same INSPECT training set examples for continued pretraining. Pretraining task labels are assigned per-CT scan and vary in density based on pretraining approach (Figure 3). Note that TTE supervision enables leveraging a patient's entire future EHR, providing 3 times more training labels on average per CT-scan over per-visit labels. TTE also captures temporal structure and time-varying disease risk, providing supervision signal that is absent in predominant SSL methods for imaging.

For evaluating our pretrained encoder, we use the frozen encoder and lightweight task head strategy outlined in Section 4. For classification tasks, we use logistic regression as the classification head (linear probe) and for TTE tasks, we employ DeepSurv (Katzman et al., 2018) as the survival head. All model search hyperparameters for pretraining and adaptation are in Appendix F.

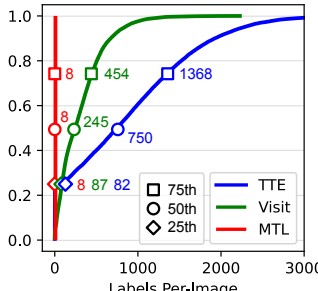

Figure 3: Label density CDF by pretraining approach.

**Metrics.** We evaluate discrimination performance using AUROC for time-thresholded binary classification tasks and Harrell's C-index (Harrell et al., 1982) for TTE tasks. Harrell's C-index is a type of C statistic that summarizes the ability of a predictive model to rank patients. Harrell's C-index requires the predictive model to output a single risk per patient, as opposed to risk over time, thus suitable for our DeepSurv head evaluation. We use the standard formulation for Harrell's C-index, where it first finds all possible pairs $P$ and one patient is known to have the event before another, then splits that set into $P_{\text{correct}}$ (the higher predicted risk patient has the event first), $P_{\text{tied}}$ (both patients are tied for the predicted risk), and $P_{\text{incorrect}}$ (lower predicted risk patient actually has the event first): $C_H = \frac{P_{\text{correct}} + 0.5 \times P_{\text{tied}}}{P_{\text{correct}} + P_{\text{tied}} + P_{\text{incorrect}}}$. Appendix D contains additional TTE evaluation metrics for time-dependent C-statistics and the integrated Brier score.

All performance results and 95% confidence intervals are reported using a test set bootstrap of $n = 1000$ replicates. Statistical significance was computed using a two-tailed Z-test (p-value at 0.05 for rejecting the null hypothesis that the difference between two sample means is zero). A complete set of statistical tests are in Appendix K.

**Additional Experiments.** See the appendix for further experimental ablations including: subgroup performance (Appendix R), task head capacity (Appendix L), and full fine-tuning vs. frozen backbone adaptation (Appendix J).

# 6 RESULTS

**Evaluating Prognostic Performance.** Table 2 reports performance for the prognostic binary classification formulations of outcome prediction for the original INSPECT tasks. This provides a simplified view of high and low-risk patients across time. We find that TTE pretrained models outperform all baselines across all architectures in our experiments. Here TTE pretraining provides an average of 22.6% performance over the `base` pretrained model and 15.4% average increase over `base/visit`. The `base/MTL` baseline also outperforms the `base` model, but underperforms TTE, highlighting the benefits of increasing pretraining tasks. Comparing `base/TTE` versus `base/visit` is more informative, as both approaches use the same number of pretraining tasks (8,192) but `base/TTE` substantially improves the density of the label per image during pretraining (Figure 3). Table 3 reports performance of all original INSPECT tasks and our five new outcomes using Harrell's C-index. Here TTE pretraining largely outperforms all of our baselines across all architectures.

| Model | Mortality | | | Readmission | | | PH |
|---|---|---|---|---|---|---|---|
| | 1M | 6M | 12M | 1M | 6M | 12M | 12M |
| SwinUNETR `base` | 0.693 | 0.684 | 0.685 | 0.507 | 0.538 | 0.569 | 0.597 |
| SwinUNETR `base/MTL` | 0.676 | 0.700 | 0.697 | 0.502 | 0.543 | 0.551 | 0.606 |
| SwinUNETR `base/visit` | 0.693 | 0.716 | 0.670 | 0.560 | 0.554 | 0.528 | 0.560 |
| SwinUNETR `base/TTE` | **0.827** | **0.808** | **0.788** | **0.582** | **0.612** | **0.607** | **0.672** |
| DenseNet-121 `base` | 0.616 | 0.568 | 0.575 | 0.512 | 0.532 | 0.524 | 0.506 |
| DenseNet-121 `base/MTL` | 0.665 | 0.596 | 0.602 | 0.533 | 0.557 | 0.547 | 0.577 |
| DenseNet-121 `base/visit` | 0.649 | 0.698 | 0.692 | 0.504 | 0.538 | 0.567 | 0.599 |
| DenseNet-121 `base/TTE` | **0.770** | **0.730** | **0.725** | **0.629** | **0.643** | **0.637** | **0.689** |
| ResNet-152 `base` | 0.583 | 0.557 | 0.554 | 0.536 | 0.537 | 0.509 | 0.537 |
| ResNet-152 `base/MTL` | 0.726 | 0.715 | 0.647 | 0.563 | 0.564 | 0.566 | 0.570 |
| ResNet-152 `base/visit` | 0.691 | 0.636 | 0.643 | 0.562 | 0.564 | 0.566 | 0.567 |
| ResNet-152 `base/TTE` | **0.804** | **0.798** | **0.792** | **0.602** | **0.649** | **0.657** | **0.718** |

Table 2: Prognostic binary classification performance for INSPECT tasks on Logistic Regression (1, 6, 12 month time horizon bins) reported as the mean AUROC of a test set bootstrap (n=1000). **Bold** indicates the best performer. Underlined indicates no statistically significant difference versus the *`base/TTE` models.

| Model | Harrell's C-Index ↑ | | | | | | | |
|---|---|---|---|---|---|---|---|---|
| | Mort. | Readm. | PH | ATX | CMG | CONS | EDM | PEFF |
| SwinUNETR $_{base}$ | 0.717 | 0.653 | 0.696 | 0.558 | 0.662 | 0.549 | 0.697 | 0.641 |
| SwinUNETR $_{base/MTL}$ | 0.672 | 0.671 | 0.665 | 0.681 | 0.677 | 0.668 | 0.715 | 0.668 |
| SwinUNETR $_{base/visit}$ | 0.671 | 0.716 | 0.697 | 0.719 | 0.717 | 0.718 | 0.717 | 0.714 |
| SwinUNETR $_{base/TTE}$ | **0.738** | **0.723** | **0.724** | **0.739** | **0.739** | **0.738** | **0.738** | **0.738** |
| DenseNet-121 $_{base}$ | 0.505 | 0.505 | 0.541 | 0.505 | 0.505 | 0.506 | 0.505 | 0.536 |
| DenseNet-121 $_{base/MTL}$ | 0.589 | 0.590 | 0.591 | 0.593 | 0.589 | 0.586 | 0.587 | 0.590 |
| DenseNet-121 $_{base/visit}$ | 0.675 | 0.675 | 0.699 | 0.675 | 0.662 | 0.676 | 0.676 | 0.675 |
| DenseNet-121 $_{base/TTE}$ | **0.732** | **0.723** | **0.726** | **0.720** | **0.711** | **0.712** | **0.725** | **0.723** |
| ResNet-152 $_{base}$ | 0.505 | 0.560 | 0.505 | 0.577 | 0.505 | 0.559 | 0.505 | 0.536 |
| ResNet-152 $_{base/MTL}$ | 0.701 | 0.686 | 0.656 | 0.663 | 0.562 | 0.656 | 0.660 | 0.572 |
| ResNet-152 $_{base/visit}$ | 0.656 | 0.702 | 0.643 | 0.703 | 0.705 | 0.703 | 0.700 | 0.716 |
| ResNet-152 $_{base/TTE}$ | **0.732** | **0.739** | **0.735** | **0.728** | **0.727** | **0.727** | **0.737** | **0.737** |

Table 3: Prognostic TTE performance for INSPECT, measured by Harrell's C-Index. **Bold** indicates the best performance. Underlined indicates no statistically significant difference versus the $*_{base/TTE}$ models.

**Evaluating Diagnostic Performance.** We are also interested in assessing how TTE continued pretraining may impact standard image classification tasks, corresponding to diagnostic labeling of medical imaging for current disease biomarkers. Table 4 outlines 8 image biomarker tasks. Here, performance of all 3D model architectures is poor, especially `base` models. For almost all diagnostic tasks, TTE pretraining performs the same (i.e., statistically indistinguishable, detailed numbers shown in Table 17) as all other tested pretraining approaches. This aligns with the intuition that visit-level labels reflect current clinical events, should encode the same level of diagnostic information as TTE pretraining. This also aligns with the small performance gains reported in prior work when tasks derived from EHR codes to supervise image models (Blankemeier et al., 2024). Note that while these tasks reflect observable pixel biomarkers present in images, there is also overlap with pixel biomarkers indicative of future risk. For example, a RV/LV ratio $\geq 1$, defined as the ratio of the right ventricular (RV) diameter to the left ventricular (LV) diameter, is indicative of increased risk of mortality (Lu et al., 2012). Here TTE pretraining yields statistically significant performance improvements across all architectures and pretraining methods.

| Model | PE | | | | | | RV/LV Ratio | |
|---|---|---|---|---|---|---|---|---|
| | Left | Cent. | Right | Chronic | Acute | Indet. | <1 | ≥1 |
| SwinUNETR $_{base}$ | 0.573 | 0.571 | 0.573 | 0.507 | 0.581 | 0.721 | 0.525 | 0.578 |
| SwinUNETR $_{base/MTL}$ | 0.583 | 0.633 | 0.591 | 0.504 | 0.594 | 0.732 | 0.517 | 0.598 |
| SwinUNETR $_{base/visit}$ | 0.545 | **0.664** | 0.562 | **0.556** | 0.595 | 0.741 | 0.521 | 0.546 |
| SwinUNETR $_{base/TTE}$ | **0.634** | 0.651 | **0.633** | 0.525 | **0.716** | **0.781** | **0.643** | **0.636** |
| DenseNet-121 $_{base}$ | 0.596 | 0.615 | 0.596 | 0.565 | 0.676 | 0.724 | 0.581 | 0.607 |
| DenseNet-121 $_{base/MTL}$ | 0.612 | 0.638 | 0.615 | 0.504 | 0.664 | 0.713 | 0.594 | 0.636 |
| DenseNet-121 $_{base/visit}$ | 0.595 | 0.633 | 0.605 | 0.571 | **0.677** | 0.748 | 0.573 | 0.615 |
| DenseNet-121 $_{base/TTE}$ | **0.647** | **0.716** | **0.644** | **0.586** | 0.665 | **0.762** | **0.623** | **0.686** |
| ResNet-152 $_{base}$ | 0.654 | 0.687 | 0.618 | 0.507 | 0.695 | 0.704 | 0.593 | 0.626 |
| ResNet-152 $_{base/MTL}$ | 0.658 | **0.693** | 0.621 | 0.506 | 0.685 | 0.703 | 0.587 | 0.642 |
| ResNet-152 $_{base/visit}$ | 0.646 | 0.665 | 0.625 | **0.548** | **0.743** | 0.718 | 0.566 | 0.665 |
| ResNet-152 $_{base/TTE}$ | **0.657** | 0.687 | **0.642** | 0.504 | 0.588 | **0.722** | **0.597** | **0.708** |

Table 4: Diagnostic binary classification performance for RSPECT, reported as the mean AUROC of a test set bootstrap (n=1000). **Bold** indicates the best performer. Underlined indicates no statistically significant difference versus the $*_{base/TTE}$ models.

# 7 DISCUSSION AND CONCLUSION

Medical images hold significant, untapped potential as sources of imaging biomarkers to predict future disease risk. However, current SSL approaches for imaging largely fail to capture the temporal dynamics of long-term disease progression, inherently capturing only static, structural information. Building on this observation, our work explores a TTE pretraining technique that directly incorporates future temporal information at scale, yielding several insights.

**Current Pretraining Struggles to Learn Prognostic Pixel Biomarkers.** Existing supervised and SSL pretraining methods consistently showed lower AUROC across prognostic tasks in our experiments. When compared against `base` models, TTE pretraining showed an average increase in AUROC of 0.128 (95% CI: [0.075, 0.158]), Harrell's C-index 0.166 (95% CI [0.138, 0.490]) over three architectures on prognostic binary classification and prognostic TTE tasks respectively. This suggests that off-the-shelf pretraining methods struggle to learn pixel biomarkers associated with future disease risk, likely due to missing temporal links to future pathologies. This underscores Huang et al. (2020)'s findings that predicting disease prognosis with high accuracy and certainty remains a challenging task. Therefore, incorporating explicit temporal supervision into the pretraining process may be essential for improving prognostic tasks' performance.

**TTE Pretraining Improves Prognostic Performance.** When compared against `base/MTL` and `base/visit`, TTE supervision improves performance on Harrell's C-index on prognostic TTE task tasks by 0.093 (95% CI [0.086, 0.554]) and, 0.038 (95% CI [0.017, 0.370]) respectively. Additionally TTE supervision does not negatively impact diagnostic binary classification for significant difference between `base` and TTE across RSPECT tasks, shown in Table 17.

**TTE Supervision Improves Training Data Efficiency.** By leveraging temporal information from the future in EHRs and censored patients, TTE pretraining increases label density to boost AUROC for prognostic binary classification tasks by 0.093 (95% CI [0.053, 0.134]) and 0.092 (95% CI [0.051, 0.135]) when compared against `base/MTL` and `base/visit` models respectively. TTE supervision increases available task labels per-training instance by 3x on average (Figure 3). This underscores the potential of TTE pretraining as a viable objective for scaling medical imaging AI, given that expert-level annotation is time-consuming, expensive, and difficult to collect (Dgani et al., 2018; Tajbakhsh et al., 2021; Aljabri et al., 2022). Increasing the data efficiency of a given training example also contributes to reducing compute costs.

**Limitations.** First, our study focuses exclusively on evaluating 3D vision encoders, which demand significant memory and computational resources—80GB memory GPUs for SwinUNETR and 40GB memory GPUs for DenseNet/ResNet architectures. Second, while our study surpasses previous work in imaging-based TTE in terms of scale, our pretraining dataset remains relatively small compared to modern, general-purpose datasets (Schuhmann et al., 2022) and recently released medical datasets (Xie et al., 2024). Furthermore, we focus exclusively on a single modality, CT scans. Expanding both the scale and diversity of the pretraining data mixtures (e.g., other imaging modalities including both 2D and 3D, historical EHR and clinical text) could enhance performance or lead to a deeper understanding of the trade-offs between different architectures. Finally, since this work focuses on evaluating encoder quality, we only evaluated frozen encoders with smaller, lightweight, supervised task heads for adaptation. Alternative adaption methods under different sample assumptions, e.g., zero/few-shot learning, may reveal different performance trade-offs.

**Conclusion.** This work presents the first empirical study of using time-to-event pretraining for 3D medical vision models. By leveraging longitudinal EHRs and defining a time-to-event pretraining objective (comprising 8,192 tasks), we embed long term outcome information into the image model during pretraining. Doing so results in an average threefold increase in training labels compared to limiting just to a patient's current EHR visit. We observe substantial improvements in prediction performance for future events, with increases of up to a 31.6% in AUROC and a 40.5% improvement in Harrell's C-index, without negatively impacting standard binary classification tasks (e.g., image labeling for diagnostic tasks). Our results reveal a clear need for pretraining datasets for 3D medical foundation models to include tasks that capture long-term temporal structure, demonstrated here through our use of time-to-event supervision. This study on the utility of using longitudinal EHR records as a supervision source for future-guided pretraining of 3D medical imaging models lays the groundwork for innovative ways to combine EHR and imaging modalities for clinical risk prediction.

ACKNOWLEDGMENTS

Research reported in this publication was supported by the National Heart, Lung, and Blood Institute of the National Institutes of Health Award R01HL155410, as well as NIH grants R01HL167974, R01HL169345, P41 EB027060; ARPA-H contract 1AYSAX0000024-01; and NIH contracts 75N92020C00008 and 75N92020C00021. Further support was provided by the Stanford Institute for Human-Centered Artificial Intelligence (HAI) and the Stanford Center for Artificial Intelligence in Medicine and Imaging (AIMI) in the form of an HAI Seed Grant and AIMI-HAI Partnership Grant. We would also like to thank the Clinical Excellence Research Center (CERC) at Stanford for their support.

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

**Security, Data Storage, and Compliance:** All authors involved in data handling have completed institutional training on HIPAA and data privacy before engaging with the data. All training data was stored in a HIPAA-compliant compute environment.

**Algorithmic Bias** Healthcare machine learning models can be susceptible to algorithmic bias, leading to unfavorable outcomes for underrepresented subgroups (Obermeyer et al., 2019). Bias mitigation in medical foundation models remains an ongoing research challenge (Pfohl et al., 2024) and is not covered in this study. However, we take two steps to mitigate risk. First, all of our continued pretrained model releases includes a Data Use Agreement (DUA) that explicitly prohibits direct medical care. Second, in line with the recommendations from Chang et al. (2022), we conduct an analysis in Appendix R to assess performance across sensitive subgroups, ensuring our pretraining technique does not unfairly disadvantage any group compared to existing methods. We evaluate performance using the AUROC (bootstrapped, n=1000) on 7 binary prognostic tasks, comparing TTE against `base`, showing that TTE pretraining does not reduce performance for sensitive groups and generally improves risk ranking across all groups.

REPRODUCIBILITY STATEMENT

The code artifact necessary for reproducing the experiments in this paper can be found in the supplemental materials as a zip file. The anonymous Github link is https://anonymous.4open.science/r/future_guided_pretraining-DA6C. Our hyperparameter search grids can be found in Appendix F. All `base` pretraining weights are publicly available as detailed in Table 10. To ensure reproducibility, all experiments use researcher accessible, public medical datasets.

# APPENDIX

## A DATA PREPROCESSING

**CT Scans.** Each CT scan is preprocessed by extracting pixel data and applying a linear transformation to Hounsfield Units (HU) using the rescale slope and intercept values from the original DICOM records. To retain fine-grained details, we ensure axial slices have a thickness between 1 mm and 3 mm. Finally, we pad and center crop the images to 224 x 224 pixels.

**Electronic Health Records.** INSPECT's EHR data is provided in the Observational Medical Outcomes Partnership (OMOP) Common Data Model (CDM) schema, a standardized framework that harmonizes healthcare data from various sources to support large-scale analysis (OHDSI, 2023). We use the Athena OHDSI Vocabularies Repository (Segert, 2023) (OMOP vocabulary version: v20240830) as our knowledge graph for generating tasks.

## B INSPECT NEW TTE TASK DEFINITIONS

We have selected a set of commonly used pulmonary disease tasks (Irvin et al., 2019), that are coded in INSPECT dataset's EHR events. We use these common tasks to benchmark the time-to-event performance of our proposed TTE formulation. Each patient is assigned time-to-event=True after CT's capture for pulmonary hypertension, atelectasis, cardiomegaly, consolidation, edema, and pleural effusion. These labels indicate either the time until the first occurrence of each condition or the time until the patient is censored. The description can be found in Figure 4 for diagnostic and prognostic labels.

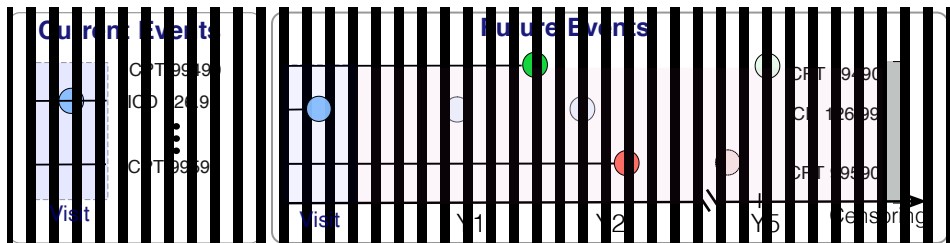

Figure 4: Overview of Label Definitions: Diagnostic tasks use labels derived from the same hospital visit as the CT scan. Prognostic tasks involve future medical events from patients' EHR timelines and are categorized into binary prognostic labels and time-to-event (TTE) prognostic labels. Note that for TTE tasks, only the time until the first occurrence is labeled.

## C CONTRIBUTION TABLE

We wish to ensure that we can accredit contributions to all the authors in a clear and fair manner and choose to follow the example from here [5].

| Author | Concept | Experiment Design | Coding | Analysis | Writing | Visualization | Project Management | Resources | Funding |
|---|---|---|---|---|---|---|---|---|---|
| Zepeng Huo* | | ✔ | ✔ | ✔ | ✔ | ✔ | ✔ | | ✔ |
| Jason Alan Fries* | ✔ | ✔ | ✔ | ✔ | ✔ | ✔ | ✔ | | ✔ |
| Alejandro Lozano* | | ✔ | ✔ | ✔ | ✔ | ✔ | | | |
| Jeya Maria Jose Valanarasu | | ✔ | ✔ | ✔ | ✔ | | | | |
| Ethan Steinberg | ✔ | ✔ | ✔ | ✔ | ✔ | | | | |
| Louis Blankemeier | | | ✔ | | | | | | |
| Akshay S. Chaudhari | | | | | | | | ✔ | |
| Curtis Langlotz | | | | | | | | ✔ | ✔ |
| Nigam H. Shah | | | | | ✔ | | ✔ | ✔ | ✔ |

Figure 5: Overview of author contributions. * denotes equal contribution.

[5] https://x.com/SteinmetzNeuro/status/1147241128858570752

# D ADDITIONAL TIME-TO-EVENT MODEL METRICS

We calculate additional metrics for time-to-event task modeling: the integrated Brier score (IBS) Graf et al. (1999), shown in Table 5, and time-dependent C-statistics Heagerty & Zheng (2005), shown in Table 6. IBS is intended to assess the discrepancy between predicted survival probabilities and observed outcomes. At each time point, the Brier score calculates the squared difference between the predicted survival probability and the actual event or censoring status. By integrating over the time range from the start to a specified maximum horizon, it summarizes the model's predictive performance over the entire period of interest. For time-dependent C-statistics, we adopt the incident and dynamic definitions for time-dependent sensitivity and specificity Heagerty & Zheng (2005) to derive time-dependent receiver operating curve $\text{AUC}(t)$. Time-dependent concordance is calculated as $C_{td} = \frac{\int_t \text{AUC}(t)w(t)dt}{\int_t w(t)dt}$, where $w(t) = f(t) \cdot S(t)$ and $f(t)$ is defined as the event rate a time $t$ and $S(t)$ is the survival probability at time $t$, both of which are estimated using Kaplan-Meier estimator Kaplan & Meier (1958). $\text{AUC}(t)$ is the integration of ROC curve under a time bin, where sensitivity and specificity are following the definitions in Heagerty & Zheng (2005). We should note that the time-dependent C-statistics is a more strict metric, especially for non-piecewise survival models, because it upweighs the long time horizon examples which are generally harder to predict for models like traditional CoxPH.

| Model | Integrated Brier Score ↓ | | | | | | | |
|---|---|---|---|---|---|---|---|---|
| | Mort. | Readm. | PH | ATX | CMG | CONS | EDM | PEFF |
| SwinUNETR base | 0.071 | 0.070 | 0.072 | 0.071 | 0.067 | 0.073 | 0.074 | 0.069 |
| SwinUNETR base/MTL | 0.081 | 0.078 | 0.079 | 0.081 | 0.078 | 0.083 | 0.081 | 0.080 |
| SwinUNETR base/visit | 0.073 | 0.068 | 0.072 | 0.070 | **0.066** | 0.073 | **0.074** | 0.070 |
| SwinUNETR base/TTE | **0.069** | **0.067** | **0.066** | **0.069** | 0.067 | **0.073** | 0.079 | **0.069** |
| DenseNet-121 base | 0.219 | 0.803 | 0.702 | 0.779 | 0.217 | 0.533 | 0.335 | 0.822 |
| DenseNet-121 base/MTL | 0.071 | 0.067 | 0.068 | 0.060 | 0.065 | 0.072 | 0.083 | 0.068 |
| DenseNet-121 base/visit | 0.063 | 0.068 | 0.071 | 0.071 | 0.060 | 0.075 | 0.075 | 0.072 |
| DenseNet-121 base/TTE | **0.028** | **0.021** | **0.022** | **0.014** | **0.014** | **0.029** | **0.023** | **0.014** |
| ResNet-152 base | 0.335 | 0.078 | 0.479 | 0.174 | 0.160 | 0.166 | 0.140 | 0.095 |
| ResNet-152 base/MTL | 0.076 | 0.074 | 0.078 | 0.075 | 0.074 | 0.077 | 0.078 | 0.077 |
| ResNet-152 base/visit | 0.075 | 0.077 | 0.074 | 0.070 | 0.067 | 0.077 | 0.074 | 0.075 |
| ResNet-152 base/TTE | **0.068** | **0.069** | **0.066** | **0.065** | **0.064** | **0.070** | **0.070** | **0.066** |

Table 5: Prognostic TTE performance for INSPECT, measured by Integrated Brier Score. **Bold** indicates the best performer. Underlined indicates no statistically significant difference versus the *base/TTE models.

| Model | Time-dependent C-Statistics ↑ | | | | | | | |
|---|---|---|---|---|---|---|---|---|
| | Mort. | Readm. | PH | ATX | CMG | CONS | EDM | PEFF |
| SwinUNETR $_{\text{base}}$ | 0.649 | 0.620 | 0.636 | 0.602 | 0.619 | 0.555 | 0.637 | 0.619 |
| SwinUNETR $_{\text{base/MTL}}$ | 0.621 | 0.624 | 0.631 | 0.619 | 0.625 | 0.631 | 0.649 | 0.633 |
| SwinUNETR $_{\text{base/visit}}$ | 0.625 | 0.645 | 0.635 | 0.650 | 0.648 | 0.644 | 0.648 | 0.638 |
| SwinUNETR $_{\text{base/TTE}}$ | **0.672** | **0.657** | **0.645** | **0.662** | **0.660** | **0.663** | **0.667** | **0.667** |
| DenseNet-121 $_{\text{base}}$ | 0.501 | 0.504 | 0.542 | 0.501 | 0.502 | 0.508 | 0.500 | 0.531 |
| DenseNet-121 $_{\text{base/MTL}}$ | 0.568 | 0.587 | 0.585 | 0.584 | 0.586 | 0.586 | 0.581 | 0.573 |
| DenseNet-121 $_{\text{base/visit}}$ | 0.632 | 0.627 | 0.637 | 0.624 | 0.618 | 0.612 | 0.624 | 0.629 |
| DenseNet-121 $_{\text{base/TTE}}$ | **0.677** | **0.657** | **0.658** | **0.651** | **0.644** | **0.651** | **0.658** | **0.657** |
| ResNet-152 $_{\text{base}}$ | 0.503 | 0.557 | 0.517 | 0.580 | 0.505 | 0.563 | 0.501 | 0.531 |
| ResNet-152 $_{\text{base/MTL}}$ | 0.632 | 0.620 | 0.621 | 0.615 | 0.604 | 0.620 | 0.618 | 0.601 |
| ResNet-152 $_{\text{base/visit}}$ | 0.622 | 0.636 | 0.621 | 0.638 | 0.631 | 0.635 | 0.640 | 0.641 |
| ResNet-152 $_{\text{base/TTE}}$ | **0.678** | **0.656** | **0.674** | **0.667** | **0.662** | **0.664** | **0.670** | **0.669** |

Table 6: Prognostic TTE performance for INSPECT, measured by time-dependent C-statistics. **Bold** indicates the best performer. Underlined indicates no statistically significant difference versus the $*_{\text{base/TTE}}$ models.

# E   MERLIN RESNET-152 RESULTS

For experimental consistency, we selected the ResNet-152 $_{base}$ model as the starting point for our continued pretraining experiments, however we also conducted an initial study on the Merlin 3D foundation model (Blankemeier et al., 2024). Merlin employs a dual-objective pretraining strategy: first, it uses contrastive loss on radiology notes associated with abdominal CT scans, followed by supervised training where EHR diagnosis codes are used as phenotype classification labels. We compare their reported best performing backbone, ResNet-152, with our weight-inflated (2D ImageNet weights) baseline ResNet-152 $_{base}$. Table 7 reports adaption performance for *prognostic classification*, *prognostic TTE* and *diagnostic classification* tasks, where ResNet-152 $_{base}$ performs similarly to Merlin. While Merlin performs better in mortality classification (though not in the TTE formulation), most tasks show no statistically significant differences across the corresponding metrics. Note that Merlin's pretraining data consists solely of abdominal CT scans, which likely limits its generalizability to a broader CT data distribution, particularly for capturing lung disease features during pretraining.

| Task Category | Task | Metric | Merlin | ResNet $_{base}$ | Best |
|---|---|---|---|---|---|
| *Prognostic Classification* | PE (0M) | | 0.518 | **0.573** | ResNet $_{base}$ |
| | Mortality (1M) | | **0.632** | 0.583 | Merlin |
| | Mortality (6M) | | **0.586** | 0.557 | Merlin |
| | Mortality (12M) | AUROC | **0.590** | 0.554 | Merlin |
| | Readmission (1M) | | **0.538** | 0.536 | - |
| | Readmission (6M) | | 0.513 | **0.537** | ResNet $_{base}$ |
| | Readmission (12M) | | **0.538** | 0.509 | - |
| | PH (12M) | | 0.514 | **0.537** | - |
| *Prognostic TTE* | Mortality | | **0.549** | 0.505 | - |
| | Readmission | | 0.545 | **0.560** | ResNet $_{base}$ |
| | PH | | **0.615** | 0.505 | - |
| | ATX | Harrell's C-index | 0.565 | **0.577** | ResNet $_{base}$ |
| | CMG | | **0.575** | 0.505 | Merlin |
| | CONS | | **0.601** | 0.559 | - |
| | EDM | | **0.601** | 0.505 | Merlin |
| | PEFF | | **0.549** | 0.536 | - |
| *Diagnostic Classification* | PE Left | | 0.603 | **0.654** | ResNet $_{base}$ |
| | PE Cent. | | 0.572 | **0.687** | ResNet $_{base}$ |
| | PE Right | | 0.562 | **0.618** | ResNet $_{base}$ |
| | PE Chronic | AUROC | **0.550** | 0.507 | - |
| | PE Acute | | 0.647 | **0.695** | - |
| | PE Indeterminate | | 0.693 | **0.704** | - |
| | RV/LV $<1$ | | 0.515 | **0.593** | ResNet $_{base}$ |
| | RV/LV $\geq 1$ | | 0.572 | **0.626** | ResNet $_{base}$ |

Table 7: Performance comparison of Merlin and ResNet $_{base}$ across various tasks. Each task is evaluated using either AUROC for binary classification tasks or Harrell's C-index for TTE tasks. **Bold** indicates the best performer. Underlined indicates no statistically significant difference between models (i.e., p>0.05). The Best column denotes the model with the highest performance for each task. Note: Merlin is pretrained on abdominal CTs.

# F  MODELING HYPERPARAMETERS

We have detailed our hyperparameter set, both in the image backbones and the probing methods, in Table 8.

| Model | Hyperparameters | Values |
|---|---|---|
| **Image backbones** | **SwinUNETR** | |
| | learning rate | $10^{-4}, 10^{-5}, 10^{-6}$ |
| | dropout prob | 0.1, 0.2, 0.3 |
| | patch size | 2x2x2 |
| | window size | 7x7x7, 8x8x8 |
| | augmentation strategies | Random rotations, flips |
| | **DenseNet** | |
| | learning rate | $10^{-3}, 10^{-4}, 10^{-5}, 10^{-6}$ |
| | depth | 121, 169 |
| | num. of dense blocks | 3,4 |
| | dropout prob | 0.1, 0.2, 0.3 |
| | augmentation strategies | Random rotations, flips |
| | **ResNet** | |
| | learning rate | $10^{-3}, 10^{-4}, 10^{-5}, 10^{-6}$ |
| | depth | 152 |
| | residual block | BottleneckBlock |
| | dropout prob | 0.1, 0.2, 0.3 |
| | augmentation strategies | Random rotations, flips |
| **Probing heads** | **Logistic Regression** | |
| | penalty | L-1, L-2 |
| | learning rate | 0.01, 0.1, 0.2 |
| | solver | lbfgs, liblinear |
| | scoring | roc_auc, accuracy |
| | software version | scikit-learn 1.3.2 |
| | **DeepSurv** | |
| | num. nodes per layer | 32, 64, 128, 256 |
| | depth | 2, 3, 4 |
| | learning rate | $10^{-4}, 10^{-5}, 10^{-6}$ |
| | dropout prob | 0.1, 0.2, 0.3 |
| | software version | pycox 0.2.3 |

Table 8: Hyperparameter search grids and software versions of methods under comparison.

# G    CLINICAL TASK SUMMARY

Since INSPECT and RSPECT both focus on patients suspected of having pulmonary embolisms, we focus on pulmonary-related pathologies in our experiments. Table 9 outlines the clinical significance of these tasks.

| Abbr. | Name | Clinical Meaning |
|---|---|---|
| PH | Pulmonary Hypertension | A condition characterized by high blood pressure in the pulmonary arteries, leading to right heart strain and potentially heart failure. Predicting on long term outcome is of higher clinical usage. |
| PE | Pulmonary Embolism | A life-threatening condition where a blood clot blocks the pulmonary arteries, leading to impaired blood flow to the lungs and acute respiratory symptoms. |
| ATX | Atelectasis | A collapse of lung tissue that prevents normal oxygen absorption, often resulting from obstruction or fluid accumulation. |
| CMG | Cardiomegaly | An abnormal enlargement of the heart, typically caused by high blood pressure or heart valve disease, which can lead to heart failure. |
| CONS | Consolidation | A region of lung tissue filled with liquid instead of air, often seen in pneumonia, causing reduced oxygen exchange. |
| EDM | Edema | The abnormal accumulation of fluid in tissues or alveolar spaces, often related to heart failure or lung injury, causing difficulty in breathing. |
| PEFF | Pleural Effusion | The buildup of excess fluid between the layers of the pleura surrounding the lungs, often due to infection, heart failure, or malignancy, leading to difficulty breathing. |

Table 9: Clinical definitions of pulmonary conditions used to define TTE tasks in the INSPECT dataset.

# H    base PRETRAINING DATASETS

The pretrained image backbone $_{base}$, i.e. SwinUNETR, DenseNet and ResNet, have their corresponding pretrained dataset before we take them in for continued pretraining. The details are described in Table 10.

| Architecture | Method | Loss | Dim. | Dataset | Size |
|---|---|---|---|---|---|
| SwinUNETR$_{base}$ | Self-supervised | MAE | 3D | Custom Medical | 10,050 |
| DenseNet-121$_{base}$ | Supervised | BCE | 2D | ImageNet | 1,281,167 |
| ResNet-152$_{base}$ | Supervised | BCE | 2D | ImageNet | 1,281,167 |

Table 10: Summary of model architectures, pretraining approaches, and off-the-shelf weights. Custom Medical includes 3D Chest, Abdomen, and Head/Neck CT and MRI scans (Valanarasu et al., 2023).

# I    EXPERIMENT RUNTIME COST

For our experiments, we have trained the models in a PHI-compliant on-premise server that uses 2 compute nodes, each with 24 Intel Xeon 2.7GHz CPU cores, 200 GB RAM and 4 Nvidia A100 GPUs or 4 H100 GPUs. The total estimate cost for our model $_{base/TTE}$ regime pretriaing is shown in Table 11.

| Architecture | Number of GPUs | Estimated wall-clock time | Estimated GPU hours |
|---|---|---|---|
| SwinUNETR$_{base/TTE}$ | 4 H100 (80GB) | 15 days | 1,440 GPU hours |
| DenseNet-121$_{base/TTE}$ | 4 A100 (40GB) | 9 days | 864 GPU hours |
| ResNet-152$_{base/TTE}$ | 4 A100 (80GB) | 10 days | 960 GPU hours |

Table 11: Summary of compute cost across architectures

## J    FULL PARAMETER FINE-TUNING ANALYSIS

To illustrate the data efficiency of TTE pretraining method, we conduct single task full parameter fine tuning on both INSPECT and RSPECT dataset, starting from $_\text{base}$ model checkpoint. This experiment is to show what is the cost to take off-the-shelf model to evaluate on the same set of tasks as ours but following traditional machine learning pipeline to conduct full parameter fine-tuning towards each task. Due to the high number of tasks, i.e. 7 binary classification tasks and 8 TTE tasks from INSPECT data, as well as 8 binary classification tasks from RSPECT data, we only choose one representative tasks from each data set and conduct the fine-tuning. We train until the model has early stopping till the plateau on validation set measured by cross entropy loss. The results are shown in Table 12. We can conclude that single task full parameter fine-tuning does not scale as well as TTE pretraining, and in general the performance is no better than, if not much worse than TTE pretraining.

| Dataset (fine-tuning task) | Architecture | Full param fine-tuned results (AUROC) | Linear probe results (AUROC) | Delta |
|---|---|---|---|---|
| INSPECT (12 month PH) | $\text{SwinUNETR}_\text{base}$ | 0.616 | 0.597 | + 3.18 % |
| | $\text{SwinUNETR}_\text{base/TTE}$ | 0.637 | 0.672 | - 5.20 % |
| | $\text{DenseNet-121}_\text{base}$ | 0.553 | 0.506 | + 9.28 % |
| | $\text{DenseNet-121}_\text{base/TTE}$ | 0.631 | 0.689 | - 8.41 % |
| | $\text{ResNet-152}_\text{base}$ | 0.552 | 0.537 | + 2.79 % |
| | $\text{ResNet-152}_\text{base/TTE}$ | 0.649 | 0.718 | - 9.61 % |
| RSPECT (RV/LV $\geq$ 1) | $\text{SwinUNETR}_\text{base}$ | 0.606 | 0.578 | + 4.84 % |
| | $\text{SwinUNETR}_\text{base/TTE}$ | 0.631 | 0.636 | - 0.79 % |
| | $\text{DenseNet-121}_\text{base}$ | 0.643 | 0.607 | + 5.93 % |
| | $\text{DenseNet-121}_\text{base/TTE}$ | 0.611 | 0.686 | - 10.93 % |
| | $\text{ResNet-152}_\text{base}$ | 0.653 | 0.626 | + 4.31 % |
| | $\text{ResNet-152}_\text{base/TTE}$ | 0.588 | 0.708 | - 16.95 % |

Table 12: Summary of compute cost across architectures for full parameter fine-tuning on one single task from each dataset. On average the full parameter fine-tuning computation cost for both INSPECT and RSPECT data ranges from 57.5 (standard deviation: 61.4) to 31.3 (standard deviation: 16.7) GPU hours respectively, which is much more expensive than linear probe, i.e. logistic regression on CPU, which takes minute level computation cost. In addition, the models when fine-tuned on single label task, tend to overfit given the large amount of parameter counts. The above results are from more stringent regularization training.

## K CONFIDENCE INTERVALS

Here we are showing the confidence interval differences between all the baselines against the proposed method ($_{\text{base}/\text{TTE}}$) in both classification tasks (measured by AUROC, in Table 13) and time-to-event tasks (measured by Harrell's C-index, in Table 14, time-dependent C-statistics, in Table 15 and integrated Brier score in Table 16). The RSPECT dataset confidence interval difference is detailed in Table 17.

| Model | Mortality | | | Readmission | | | PH |
|---|---|---|---|---|---|---|---|
| | 1M | 6M | 12M | 1M | 6M | 12M | 12M |
| SwinUNETR $_{\text{base}}$ | (-0.151, -0.055)* | (-0.156, -0.082)* | (-0.193, -0.113)* | (-0.153, -0.011)* | (-0.093, -0.001)* | (-0.072, -0.011)* | (-0.084, -0.005)* |
| SwinUNETR $_{\text{base}/\text{MTL}}$ | (-0.103, -0.023)* | (-0.123, -0.053)* | (-0.112, -0.042)* | (-0.140, -0.013)* | (-0.093, -0.002)* | (-0.074, -0.005)* | (-0.083, -0.001)* |
| SwinUNETR $_{\text{base}/\text{visit}}$ | (-0.123, -0.034)* | (-0.141, -0.064)* | (-0.124, -0.055)* | (-0.135, 0.005) | (-0.092, -0.004)* | (-0.075, -0.004)* | (-0.085, -0.011)* |
| SwinUNETR $_{\text{base}/\text{TTE}}$ | (0,0) | (0,0) | (0,0) | (0,0) | (0,0) | (0,0) | (0,0) |
| DenseNet-121 $_{\text{base}}$ | (-0.193, -0.115)* | (-0.155, -0.092)* | (-0.173, -0.114)* | (-0.185, -0.096)* | (-0.114, -0.044)* | (-0.123, -0.063)* | (-0.161, -0.094)* |
| DenseNet-121 $_{\text{base}/\text{MTL}}$ | (-0.195, -0.113)* | (-0.153, -0.095)* | (-0.171, -0.113)* | (-0.194, -0.094)* | (-0.112, -0.044)* | (-0.123, -0.065)* | (-0.162, -0.092)* |
| DenseNet-121 $_{\text{base}/\text{visit}}$ | (-0.155, -0.062)* | (-0.193, -0.133)* | (-0.193, -0.135)* | (-0.195, -0.095)* | (-0.145, -0.065)* | (-0.154, -0.076)* | (-0.163, -0.093)* |
| DenseNet-121 $_{\text{base}/\text{TTE}}$ | (0,0) | (0,0) | (0,0) | (0,0) | (0,0) | (0,0) | (0,0) |
| ResNet-152 $_{\text{base}}$ | (-0.235, -0.143)* | (-0.257, -0.172)* | (-0.222, -0.164)* | (-0.115, -0.006)* | (-0.116, -0.022)* | (-0.102, -0.016)* | (-0.234, -0.152)* |
| ResNet-152 $_{\text{base}/\text{MTL}}$ | (-0.174, -0.094)* | (-0.155, -0.085)* | (-0.145, -0.083)* | (-0.093, 0.054) | (-0.113, -0.022)* | (-0.112, -0.025)* | (-0.171, -0.094)* |
| ResNet-152 $_{\text{base}/\text{visit}}$ | (-0.132, -0.056)* | (-0.123, -0.063)* | (-0.113, -0.056)* | (-0.096, -0.062)* | (-0.132, -0.032)* | (-0.117, -0.037)* | (-0.113, -0.042)* |
| ResNet-152 $_{\text{base}/\text{TTE}}$ | (0,0) | (0,0) | (0,0) | (0,0) | (0,0) | (0,0) | (0,0) |

Table 13: 95% confidence interval **differences** for classification performance on INSPECT dataset for proposed method ($_{\text{base}/\text{TTE}}$) and baselines, measure by AUROC. The * indicates statistical significance under p-value at 0.05 for null hypothesis. (PE: pulmonary embolism; PH: pulmonary hypertension)

| Model | Harrell's C-Index ↑ | | | | | | | |
|---|---|---|---|---|---|---|---|---|
| | Mort. | Readm. | PH | ATX | CMG | CONS | EDM | PEFF |
| SwinUNETR $_{\text{base}}$ | (-0.381, -0.178)* | (-0.615, -0.126)* | (-0.925, -0.134)* | (-0.292, -0.072)* | (-0.210, -0.121)* | (-0.272, -0.164)* | (-0.455, -0.190)* | (-0.630, -0.264)* |
| SwinUNETR $_{\text{base}/\text{MTL}}$ | (-0.726, -0.061)* | (-0.404, -0.065)* | (-0.837, -0.138)* | (-0.162, -0.043)* | (-0.603, -0.002)* | (-0.897, -0.072)* | (-0.608, -0.017)* | (-0.086, -0.080)* |
| SwinUNETR $_{\text{base}/\text{visit}}$ | (-0.104, -0.011)* | (-0.441, 0.083) | (-0.699, -0.077)* | (-0.184, -0.001)* | (-0.333, -0.078)* | (-0.214, -0.127)* | (-0.605, 0.090) | (-0.776, -0.053)* |
| SwinUNETR $_{\text{base}/\text{TTE}}$ | (0, 0) | (0, 0) | (0, 0) | (0, 0) | (0, 0) | (0, 0) | (0, 0) | (0, 0) |
| DenseNet-121 $_{\text{base}}$ | (-0.564, -0.063)* | (-0.869, -0.067)* | (-0.805, -0.143)* | (-0.235, -0.206)* | (-0.495, -0.298)* | (-0.603, -0.133)* | (-0.672, -0.166)* | (-0.365, -0.053)* |
| DenseNet-121 $_{\text{base}/\text{MTL}}$ | (-0.155, -0.073)* | (-0.579 0.001)* | (-0.596, -0.146)* | (-0.663, -0.120)* | (-0.644, -0.047)* | (-0.576, -0.003)* | (-0.965, -0.348)* | (-0.829, -0.141)* |
| DenseNet-121 $_{\text{base}/\text{visit}}$ | (-0.340, -0.134)* | (-0.642, -0.108)* | (-0.189, 0.044) | (-0.463, -0.053)* | (-0.672, -0.079)* | (-0.420, -0.058)* | (-0.709, -0.071)* | (-0.273, -0.086)* |
| DenseNet-121 $_{\text{base}/\text{TTE}}$ | (0, 0) | (0, 0) | (0, 0) | (0, 0) | (0, 0) | (0, 0) | (0, 0) | (0, 0) |
| ResNet-152 $_{\text{base}}$ | (-0.332, -0.164)* | (-0.443, -0.257)* | (-0.275, -0.030)* | (-0.615, -0.228)* | (-0.518, -0.016)* | (-0.332, -0.041)* | (-0.289, -0.023)* | (-0.574, -0.165)* |
| ResNet-152 $_{\text{base}/\text{MTL}}$ | (-0.225, 0.177) | (-0.131, -0.098)* | (-0.440, -0.047)* | (-0.560, -0.284)* | (-0.834, -0.155)* | (0.568, 0.008)* | (-0.672, -0.273)* | (-0.427, -0.041)* |
| ResNet-152 $_{\text{base}/\text{visit}}$ | (-0.247, -0.166)* | (-0.541, -0.098)* | (-0.178, -0.012)* | (-0.250, -0.019)* | (-0.227, -0.080)* | (-0.186, -0.086)* | (-0.122, -0.056)* | (-0.055, 0.832) |
| ResNet-152 $_{\text{base}/\text{TTE}}$ | (0, 0) | (0, 0) | (0, 0) | (0, 0) | (0, 0) | (0, 0) | (0, 0) | (0, 0) |

Table 14: 95% confidence interval **differences** for time-to-event performance on INSPECT dataset for proposed method ($_{\text{base}/\text{TTE}}$) and baselines, measured by Harrell's C-Index. The * indicates statistical significance under p-value at 0.05 for null hypothesis.

| Model | Time-dependent c-statistics ↑ | | | | | | | |
|---|---|---|---|---|---|---|---|---|
| | Mort. | Readm. | PH | ATX | CMG | CONS | EDM | PEFF |
| SwinUNETR $_{base}$ | $(-0.051, -0.009)*$ | $(-0.050, -0.010)*$ | $(-0.044, -0.016)*$ | $(-0.107, -0.049)*$ | $(-0.074, -0.014)*$ | $(-0.132, -0.072)*$ | $(-0.060, -0.010)*$ | $(-0.090, -0.030)*$ |
| SwinUNETR $_{base/MTL}$ | $(-0.053, -0.007)*$ | $(-0.055, -0.005)*$ | $(-0.053, -0.005)*$ | $(-0.073, -0.013)*$ | $(-0.050, -0.020)*$ | $(-0.062, -0.032)*$ | $(-0.042, -0.012)*$ | $(-0.049, -0.019)*$ |
| SwinUNETR $_{base/visit}$ | $(-0.040, -0.020)*$ | $(-0.037, -0.023)*$ | $(-0.045, -0.015)*$ | $(-0.052, -0.002)*$ | $(-0.049, -0.019)*$ | $(-0.048, -0.018)*$ | $(-0.034, -0.004)*$ | $(-0.033, -0.003)*$ |
| SwinUNETR $_{base/TTE}$ | $(0, 0)$ | $(0, 0)$ | $(0, 0)$ | $(0, 0)$ | $(0, 0)$ | $(0, 0)$ | $(0, 0)$ | $(0, 0)$ |
| DenseNet-121 $_{base}$ | $(-0.213, -0.143)*$ | $(-0.222, -0.152)*$ | $(-0.146, -0.126)*$ | $(-0.194, -0.144)*$ | $(-0.185, -0.135)*$ | $(-0.163, -0.123)*$ | $(-0.189, -0.149)*$ | $(-0.181, -0.131)*$ |
| DenseNet-121 $_{base/MTL}$ | $(-0.109, -0.069)*$ | $(-0.100, -0.070)*$ | $(-0.093, -0.063)*$ | $(-0.090, -0.060)*$ | $(-0.082, -0.052)*$ | $(-0.095, -0.065)*$ | $(-0.093, -0.063)*$ | $(-0.094, -0.064)*$ |
| DenseNet-121 $_{base/visit}$ | $(-0.075, -0.045)*$ | $(-0.060, -0.030)*$ | $(-0.051, -0.021)*$ | $(-0.063, -0.033)*$ | $(-0.050, -0.020)*$ | $(-0.060, -0.030)*$ | $(-0.064, -0.034)*$ | $(-0.058, -0.028)*$ |
| DenseNet-121 $_{base/TTE}$ | $(0, 0)$ | $(0, 0)$ | $(0, 0)$ | $(0, 0)$ | $(0, 0)$ | $(0, 0)$ | $(0, 0)$ | $(0, 0)$ |
| ResNet-152 $_{base}$ | $(-0.258, -0.188)*$ | $(-0.099, -0.069)*$ | $(-0.172, -0.142)*$ | $(-0.102, -0.072)*$ | $(-0.247, -0.217)*$ | $(-0.106, -0.076)*$ | $(-0.192, -0.162)*$ | $(-0.188, -0.158)*$ |
| ResNet-152 $_{base/MTL}$ | $(-0.067, -0.037)*$ | $(-0.086, -0.056)*$ | $(-0.063, -0.033)*$ | $(-0.067, -0.037)*$ | $(-0.077, -0.047)*$ | $(-0.059, -0.029)*$ | $(-0.061, -0.031)*$ | $(-0.078, -0.048)*$ |
| ResNet-152 $_{base/visit}$ | $(-0.062, -0.032)*$ | $(-0.035, -0.005)*$ | $(-0.063, -0.033)*$ | $(-0.059, -0.029)*$ | $(-0.046, -0.016)*$ | $(-0.044, -0.014)*$ | $(-0.029, 0.001)$ | $(-0.028, -0.002)*$ |
| DenseNet-121 $_{base/TTE}$ | $(0, 0)$ | $(0, 0)$ | $(0, 0)$ | $(0, 0)$ | $(0, 0)$ | $(0, 0)$ | $(0, 0)$ | $(0, 0)$ |

Table 15: 95% confidence interval **differences** for time-to-event performance on INSPECT dataset for proposed method ($_{base/TTE}$) and baselines, measured by time-dependent C-statistics. The * indicates statistical significance under $p$-value at 0.05 for null hypothesis.

| Model | Integrated Brier Score ↓ | | | | | | | |
|---|---|---|---|---|---|---|---|---|
| | Mort. | Readm. | PH | ATX | CMG | CONS | EDM | PEFF |
| SwinUNETR $_{base}$ | $(0.004, 0.016)*$ | $(0.006, 0.014)*$ | $(0.005, 0.017)*$ | $(0.015, 0.035)*$ | $(0.010, 0.030)*$ | $(0.021, 0.041)*$ | $0.020 (0.010, 0.030)*$ | $(0.012, 0.032)*$ |
| SwinUNETR $_{base/MTL}$ | $(0.002, 0.022)*$ | $(0.001, 0.021)*$ | $(0.003, 0.023)*$ | $(0.003, 0.027)*$ | $(-0.004, 0.026)$ | $(0.005, 0.025)*$ | $(0.013, 0.017)*$ | $(-0.004, 0.026)$ |
| SwinUNETR $_{base/visit}$ | $(-0.006, 0.014)$ | $(0.009, 0.011)*$ | $(0.005, 0.017)*$ | $(0.014, 0.016)*$ | $(0.016, 0.014)*$ | $(0.015, 0.015)*$ | $(-0.020, 0.010)$ | $(0.014, 0.016)*$ |
| SwinUNETR $_{base/TTE}$ | $(0, 0)$ | $(0, 0)$ | $(0, 0)$ | $(0, 0)$ | $(0, 0)$ | $(0, 0)$ | $(0, 0)$ | $(0, 0)$ |
| DenseNet-121 $_{base}$ | $(0.176, 0.206)*$ | $(0.736, 0.786)*$ | $(0.663, 0.697)*$ | $(0.751, 0.779)*$ | $(0.188, 0.218)*$ | $(0.485, 0.523)*$ | $(0.295, 0.329)*$ | $(0.794, 0.822)*$ |
| DenseNet-121 $_{base/MTL}$ | $(0.033, 0.053)*$ | $(0.031, 0.061)*$ | $(0.029, 0.063)*$ | $(0.039, 0.069)*$ | $(0.036, 0.066)*$ | $(0.024, 0.062)*$ | $(0.041, 0.079)*$ | $(0.039, 0.069)*$ |
| DenseNet-121 $_{base/visit}$ | $(0.032, 0.062)*$ | $(0.032, 0.062)*$ | $(0.032, 0.066)*$ | $(0.042, 0.072)*$ | $(0.039, 0.069)*$ | $(0.027, 0.065)*$ | $(0.033, 0.071)*$ | $(0.043, 0.073)*$ |
| DenseNet-121 $_{base/TTE}$ | $(0, 0)$ | $(0, 0)$ | $(0, 0)$ | $(0, 0)$ | $(0, 0)$ | $(0, 0)$ | $(0, 0)$ | $(0, 0)$ |
| ResNet-152 $_{base}$ | $(0.266, 0.296)*$ | $(0.006, 0.024)*$ | $(0.412, 0.442)*$ | $(0.094, 0.124)*$ | $(0.086, 0.116)*$ | $(0.081, 0.111)*$ | $(0.055, 0.085)*$ | $(0.028, 0.058)*$ |
| ResNet-152 $_{base/MTL}$ | $(0.001, 0.019)*$ | $(0.005, 0.015)*$ | $(0.003, 0.027)*$ | $(0.005, 0.025)*$ | $(0.005, 0.025)*$ | $(0.008, 0.022)*$ | $(0.007, 0.023)*$ | $(-0.004, 0.026)$ |
| ResNet-152 $_{base/visit}$ | $(0.003, 0.017)*$ | $(0.002, 0.018)*$ | $(0.007, 0.023)*$ | $(0.010, 0.020)*$ | $(0.012, 0.018)*$ | $(0.008, 0.022)*$ | $(0.011, 0.019)*$ | $(0.010, 0.020)*$ |
| ResNet-152 $_{base/TTE}$ | $(0, 0)$ | $(0, 0)$ | $(0, 0)$ | $(0, 0)$ | $(0, 0)$ | $(0, 0)$ | $(0, 0)$ | $(0, 0)$ |

Table 16: 95% confidence interval **differences** time-to-event performance on the INSPECT dataset for the proposed method ($_{base/TTE}$) and baselines, measured by integrated Brier score. The * indicates statistical significance under p-value at 0.05 for null hypothesis.

| Model | PE | | | | | | RV/LV Ratio | |
|---|---|---|---|---|---|---|---|---|
| | Left | Cent. | Right | Chronic | Acute | Indet. | <1 | ≥1 |
| SwinUNETR $_{base}$ | (-0.094, -0.029)* | (-0.139, -0.012)* | (-0.089, -0.035)* | (-0.159, 0.108) | (-0.274, -0.010)* | (-0.131, -0.006)* | (-0.160, -0.075)* | (-0.092, -0.021)* |
| SwinUNETR $_{base/MTL}$ | (-0.097, -0.006)* | (-0.085, 0.058) | (-0.085, -0.005)* | (-0.120, 0.178) | (-0.208, -0.030)* | (-0.158, 0.060) | (-0.174, -0.071)* | (-0.084, -0.009)* |
| SwinUNETR $_{base/visit}$ | (-0.156, -0.017)* | (-0.096, 0.115) | (-0.122, 0.011) | (-0.120, 0.178) | (-0.290, 0.127) | (-0.206, 0.144) | (-0.203, -0.047)* | (-0.210, -0.031)* |
| SwinUNETR $_{base/TTE}$ | (0, 0) | (0, 0) | (0, 0) | (0, 0) | (0, 0) | (0, 0) | (0, 0) | (0, 0) |
| DenseNet-121 $_{base}$ | (-0.093, -0.012)* | (-0.170, -0.017)* | (-0.077, -0.011)* | (-0.098, 0.052) | (-0.046, 0.073) | (-0.129, 0.054) | (-0.082, 0.002) | (-0.132, -0.031)* |
| DenseNet-121 $_{base/MTL}$ | (-0.049, -0.024)* | (-0.119, -0.042)* | (-0.036, -0.016)* | (-0.146, -0.019)* | (-0.015, 0.018) | (-0.082, -0.010)* | (-0.042, -0.013)* | (-0.068, -0.032)* |
| DenseNet-121 $_{base/visit}$ | (-0.091, -0.013)* | (-0.155, -0.009)* | (-0.071, -0.005)* | (-0.088, 0.054) | (-0.040, 0.079) | (-0.107, 0.085) | (-0.093, -0.007)* | (-0.119, -0.018)* |
| DenseNet-121 $_{base/TTE}$ | (0, 0) | (0, 0) | (0, 0) | (0, 0) | (0, 0) | (0, 0) | (0, 0) | (0, 0) |
| ResNet-152 $_{base}$ | (-0.039, 0.034) | (-0.061, 0.058) | (-0.059, 0.015) | (-0.032, 0.043) | (-0.026, 0.233) | (-0.116, 0.068) | (-0.048, 0.038) | (-0.142, -0.026)* |
| ResNet-152 $_{base/MTL}$ | (-0.040, 0.037) | (-0.061, 0.066) | (-0.057, 0.015) | (-0.034, 0.041) | (-0.012, 0.213) | (-0.120, 0.076) | (-0.055, 0.036) | (-0.122, -0.012)* |
| ResNet-152 $_{base/visit}$ | (-0.021, -0.005)* | (-0.038, -0.012)* | (-0.024, -0.010)* | (0.030, 0.062)* | (0.012, 0.283)* | (-0.010, 0.006) | (-0.062, -0.001)* | (-0.075, -0.018)* |
| ResNet-152 $_{base/TTE}$ | (0, 0) | (0, 0) | (0, 0) | (0, 0) | (0, 0) | (0, 0) | (0, 0) | (0, 0) |

Table 17: 95% confidence interval **differences** classification performance of different methods on RSPECT dataset for diagnosis labels, measured by AUROC. * indicates the statistical significance under p value = 0.05

## L  TASK HEAD CAPACITY ON RANDOMLY INITIALIZED BACKBONES

We are interested in knowing the different probing method's utility, with random initialization from different model architectures (i.e. image backbone provides no predictive power thus only relying on probing methods), how much does probe methods provide for prediction, and what is delta of each model from that initialization. The comparisons are shown in Figures 6, 7, 8.

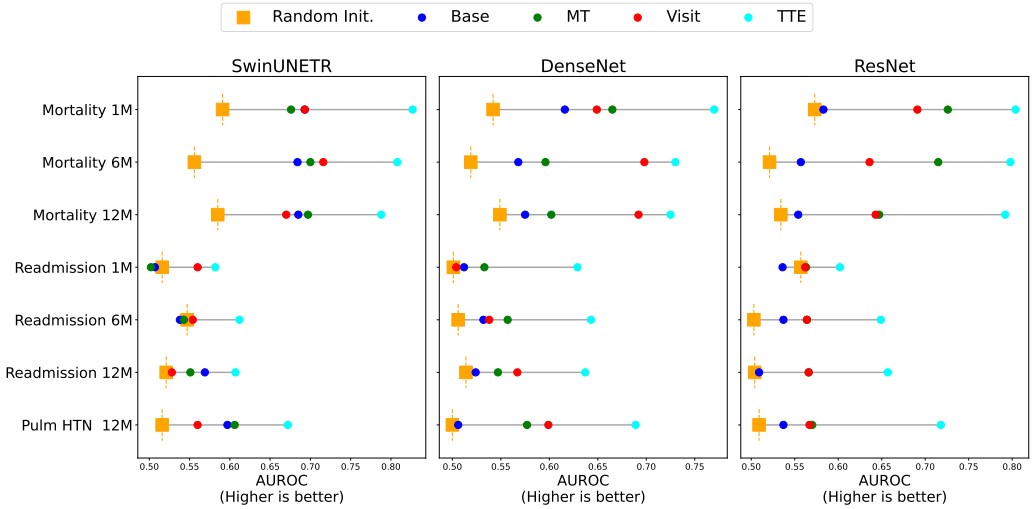

Figure 6: Comparison of models towards random initialization and each model's delta on AUROC

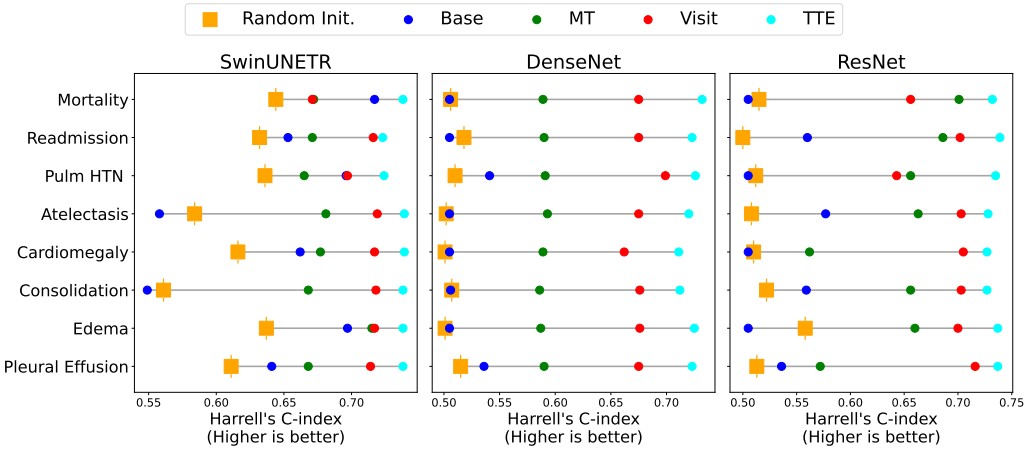

Figure 7: Comparison of models towards random initialization and each model's delta on Harrell's C-index

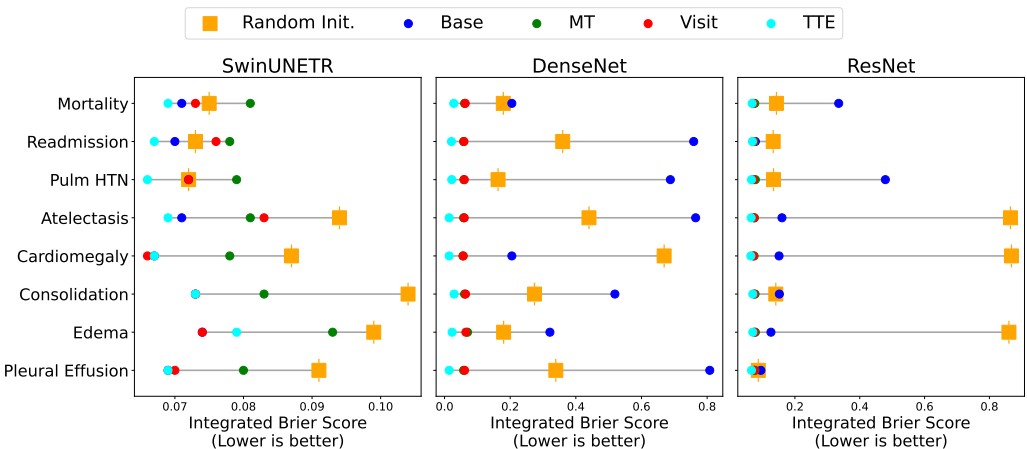

Figure 8: Comparison of models towards random initialization and each model's delta on integrated Brier score

## M  PIECEWISE EXPONENTIAL LOSS FUNCTION

In our methods section, we briefly describe that we use a piecewise exponential model, and thus, we use the corresponding piecewise exponential survival loss as our loss function. In this appendix section, we provide the exact formulas used to implement that survival loss.

For the piecewise exponential, we select a number of pieces and each piece covers the response period starting at $S_p$ and ending at $E_p$. For every piece $p$ and image $i$ in our dataset, our model needs to output $\lambda_{ip}$, the instantaneous hazard rate for the patient described by that image during the response time period $p$. These $\lambda$ values are then used to define a survival function for each image $i$:

$$S_i(t) = \prod_{p=1}^{P} \exp\left[-\lambda_{ip}(\min(t, E_p) - S_p))I(t \geq S_p)\right], \tag{3}$$

This survival function has an associated PDF, $f$, that is simply the negative of the derivative of the survival function.

$$f_i = -\frac{\partial S_i}{\partial t} \tag{4}$$

Applying the derivative operator, we obtain that

$$f_i = S_i(t) \sum_{p=1}^{P} \lambda_{ip} I(S_e \leq t \leq S_p) \tag{5}$$

We define the loss function for training by using the survival function to implement the standard survival likelihood equation and taking the product of that likelihood over the entire dataset. $\Delta_i$ is event indicator whose value is 1 when the actual event is observed. We can then plug in our definitions for the survival function $S$ and the pdf $f$ to obtain our final loss function:

$$\begin{aligned}
\mathcal{L} &= \prod_{i=i}^{n} [S_i(t)]^{1-\Delta_i} \left[f_i(t)\right]^{\delta_i} \\
\mathcal{L} &= \prod_{i=i}^{n} [S_i(t)] \left[\sum_{p=1}^{P} \lambda_{ip} I(S_e \leq t \leq S_p)\right]^{\Delta_i} \\
\mathcal{L} &= \prod_{i=i}^{n} [\prod_{p=1}^{P} \exp\left[-\lambda_{ip}(\min(t, E_p) - S_p))I(t \geq S_p)\right]] \left[\sum_{p=1}^{P} \lambda_{ip} I(S_e \leq t \leq S_p)\right]^{\Delta_i}
\end{aligned} \tag{6}$$

# N   KAPLAN-MEIER CURVES FOR TTE TASKS

We here plot the stratified groups in terms of the diagnosis of pulmonary embolism for Kaplan-Meier curves among all of our TTE tasks, shown in Figures 9, 10, 11, 12, 13, 14, 15, 16.

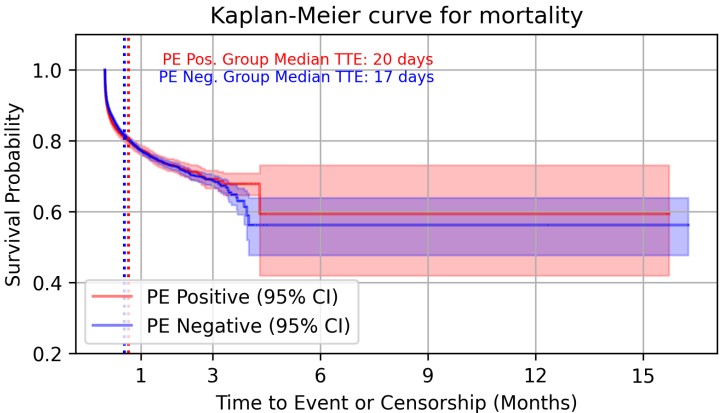

Figure 9: Kaplan-Meier curve for Mortality

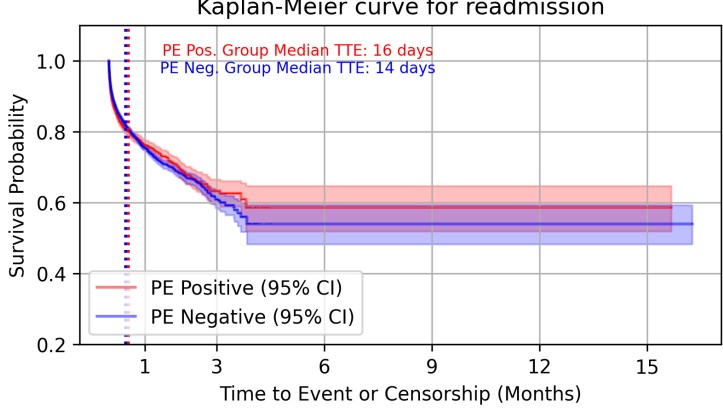

Figure 10: Kaplan-Meier curve for Readmission

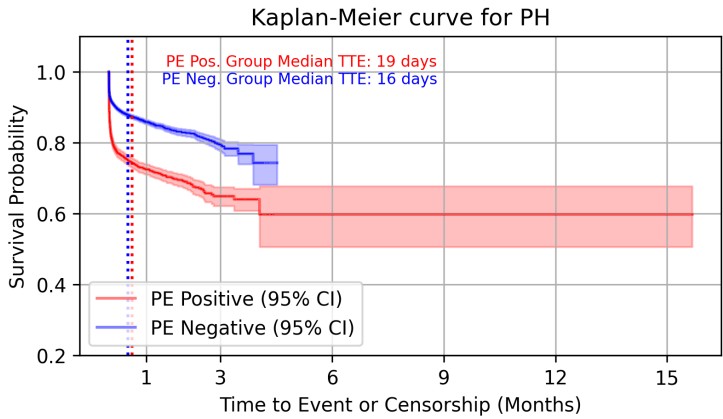

Figure 11: Kaplan-Meier curve for Pulmonary Hypertension (PH)

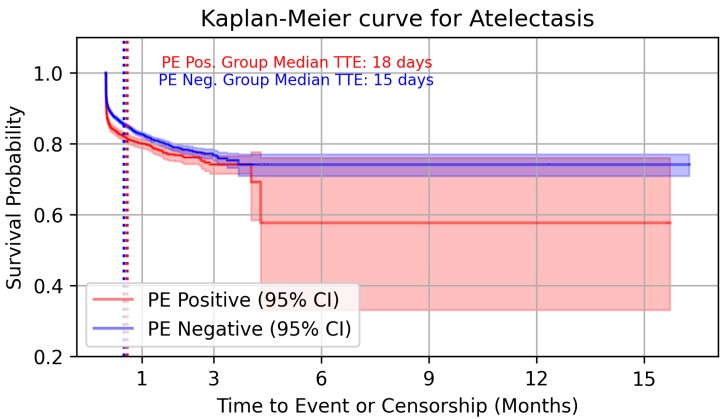

Figure 12: Kaplan-Meier curve for Atelectasis

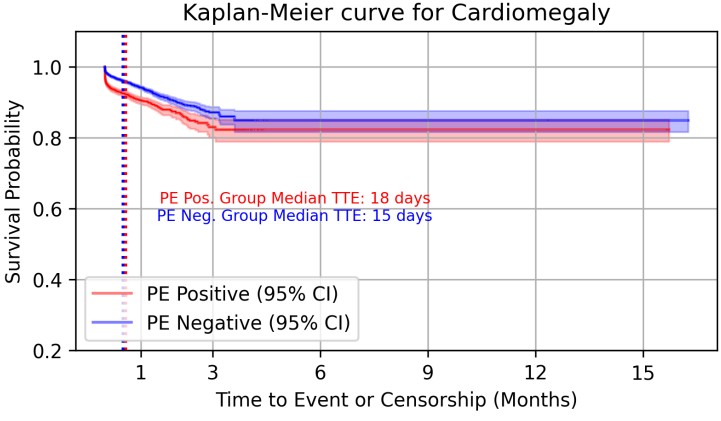

Figure 13: Kaplan-Meier curve for Cardiomegaly

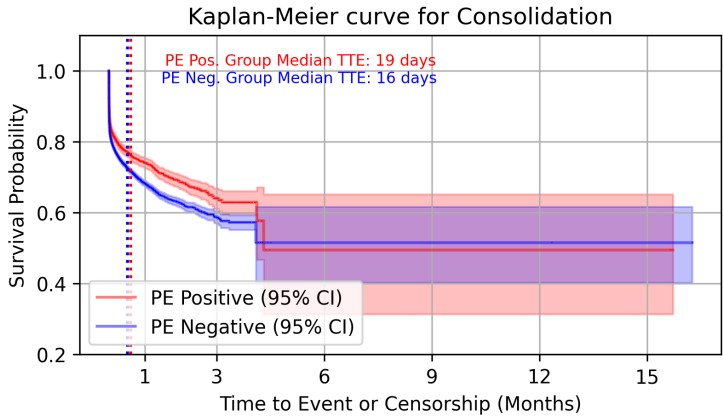

Figure 14: Kaplan-Meier curve for Consolidation

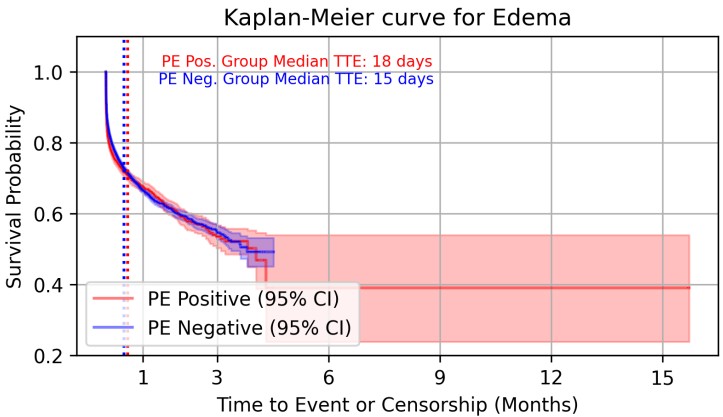

Figure 15: Kaplan-Meier curve for Edema

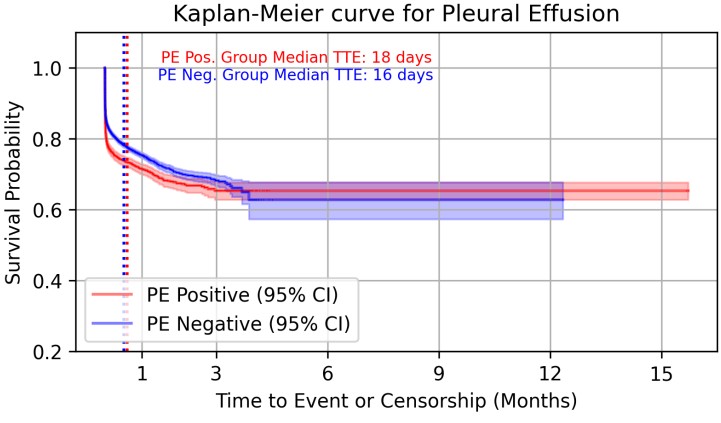

Figure 16: Kaplan-Meier curve for Pleural Effusion

## O    PRETRAINING TASK EVENT TIME DISTRIBUTIONS

We have employed 8,192 future clinical events in our pretraining mixture and here we plot the distribution of the time to event w.r.t to each of the 8,192 events across all the cohorts in INSPECT data. We have maximum and median time to event distributions for the 8192 labels and all time to event distribution for each occurrence in Figures 17, 18, 19.

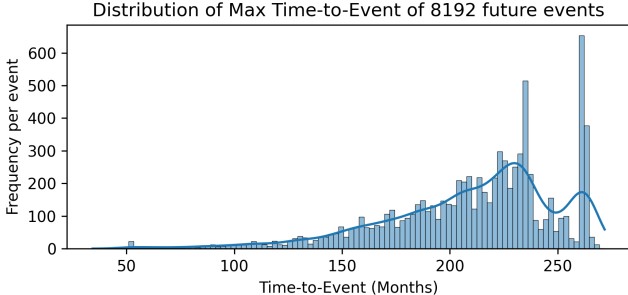

Figure 17: Maximum Time to Event across INSPECT cohort for 8192 future events (Per event)

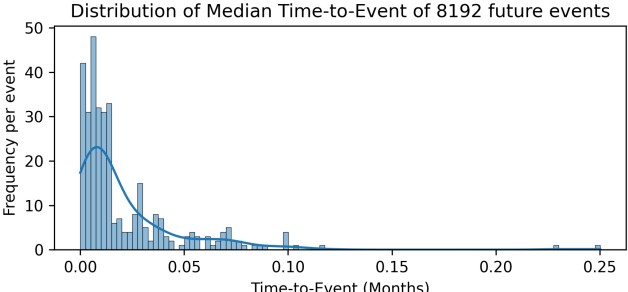

Figure 18: Median Time to Event across INSPECT cohort for 8192 future events (Per event)

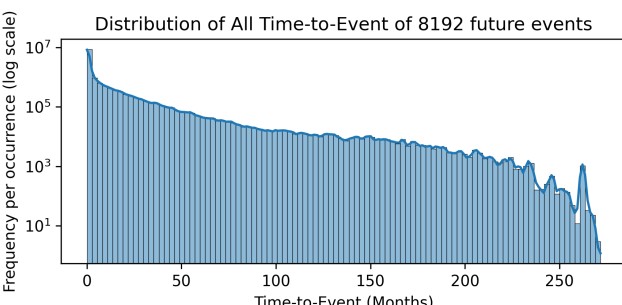

Figure 19: All Time to Event across INSPECT cohort for 8192 future events (Per occurrence)

## P    EXAMPLES OF PRETRAINING CLINICAL EVENTS

We have list the set of medical ontologies that are availabe in the EHR modality, in Table 18. We further stratify the frequency of our 8,192 pretraining clinical events across all cohorts in the INSPECT dataset into quintiles and present the top 3 examples in each quintile, in Table 19.

| Ontology |
| --- |
| OMOP Extension |
| Medicare Specialty |
| CPT4 |
| CVX |
| ICD9Proc |
| RxNorm |
| SNOMED |
| RxNorm Extension |
| Cancer Modifier |
| ICD10PCS |
| CMS Place of Service |
| Visit |
| Ethnicity |
| Gender |
| ICDO3 |
| Race |
| LOINC |
| HCPCS |

Table 18: OHDSI Athena ontolgies used in our benchmark

| Medical Code | Description | Quintile |
| --- | --- | --- |
| SNOMED/60853003 | Disorder of magnesium metabolism | 1 |
| SNOMED/167321001 | Urine dipstick for urobilinogen | 1 |
| LOINC/2472-9 | Immunoglobulin M (IgM) in serum or plasma | 1 |
| RxNorm/905216 | Hydralazine Hydrochloride 50 MG | 2 |
| SNOMED/276319003 | Finding of menstrual bleeding | 2 |
| RxNorm/979463 | Losartan potassium 100 MG | 2 |
| SNOMED/89659001 | Amylase measurement, serum | 3 |
| SNOMED/400166009 | Acquired keratoderma | 3 |
| SNOMED/70209001 | Late effect of complications of procedure (disorder) | 3 |
| LOINC/32355-0 | Bacteria identified in Specimen by Respiratory culture | 4 |
| SNOMED/56675007 | Acute heart failure | 4 |
| SNOMED/297971001 | Finding of sensation of skin | 4 |
| SNOMED/118664000 | Procedure on body system | 5 |
| SNOMED/118717007 | Procedure on organ | 5 |
| SNOMED/118672003 | Procedure on cardiovascular system | 5 |

Table 19: Top 3 events in each quintile

## Q    FEATURE ATTRIBUTION

We use GradCAM (Selvaraju et al., 2017) to visualize the Gradient-weighted Class Activation Mapping for our DenseNet-121model and its baselines, w.r.t. to the TTE tasks we curated, where we selected the CTs with the corresponding TTE labels that actually happened for the patient after the CT capture, in Figure 20. We can observe that the TTE version of the model can focus the pathology in a more reasonable region rather than scattered features learned by different baselines.

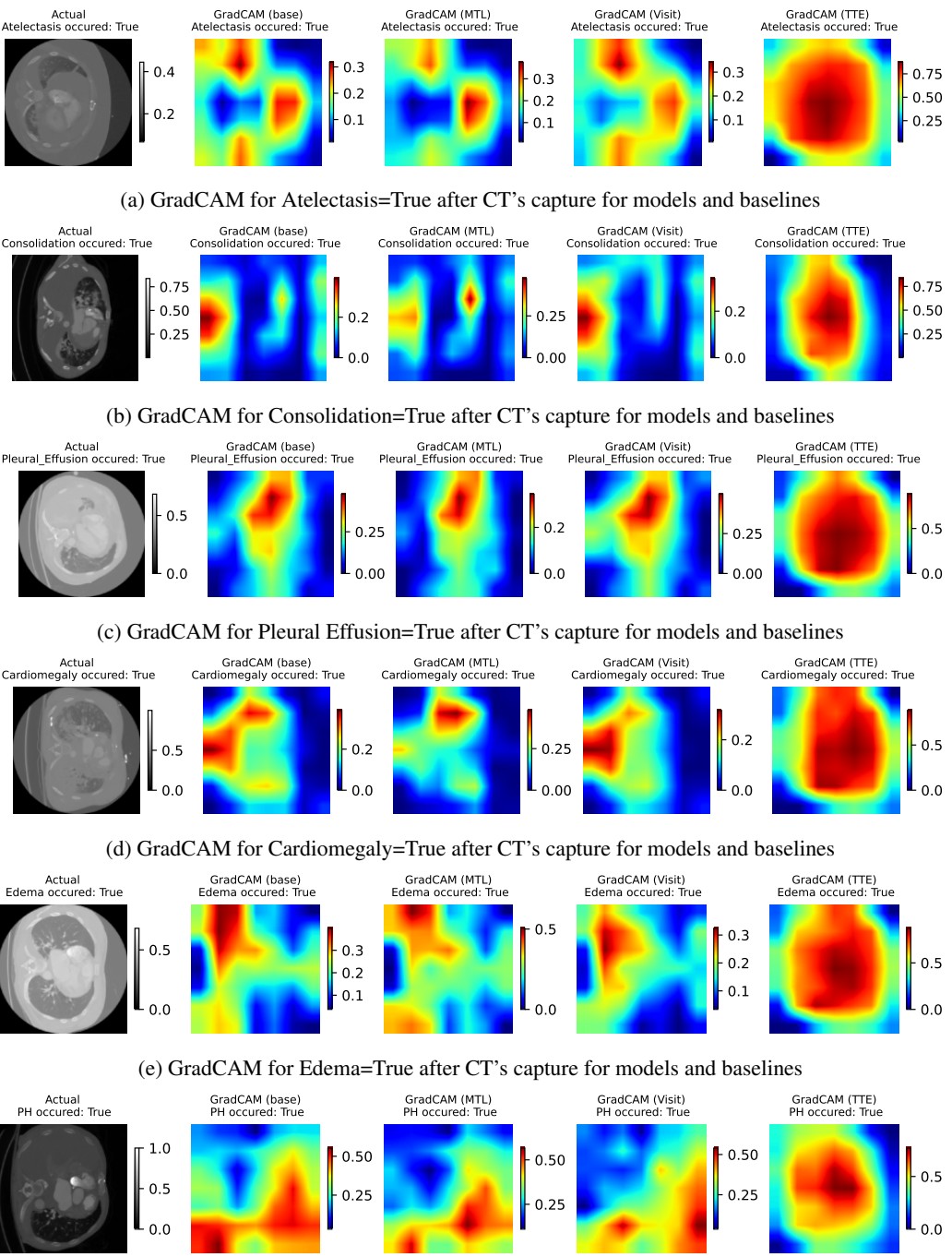

(a) GradCAM for Atelectasis=True after CT's capture for models and baselines

(b) GradCAM for Consolidation=True after CT's capture for models and baselines

(c) GradCAM for Pleural Effusion=True after CT's capture for models and baselines

(d) GradCAM for Cardiomegaly=True after CT's capture for models and baselines

(e) GradCAM for Edema=True after CT's capture for models and baselines

(f) GradCAM for Pulmonary Hypertension=True after CT's capture for models and baselines

Figure 20: GradCAM visualizations of sampled CTs for various occurred conditions: Atelectasis, Consolidation, Pleural Effusion, Cardiomegaly, Edema, and Pulmonary Hypertension.

## R    Subgroup Performance

We conduct an analysis to assess performance on our best performing model, SwinUNETR, across sensitive subgroups, ensuring our pretraining technique does not unfairly disadvantage any group compared to existing methods. The stratified subgroup counts for this experiment under test split of INSPECT dataset is in Table 20. We evaluate performance using the AUROC (bootstrapped, n=1000) on 5 binary prognostic tasks, comparing TTE against `base`, showing that TTE pretraining does not reduce performance for sensitive groups and generally improves risk ranking across all groups. The details are in Tables 21, 22, 23, 24, 25, 26, 27.

| Group | Concept | Counts |
|---|---|---|
| Gender | Female | 1844 |
| | Male | 1370 |
| Age | 18-39 | 448 |
| | 39-69 | 1667 |
| | 69-89 | 993 |
| | >89 | 106 |
| Race | Asian | 554 |
| | Black | 192 |
| | Native | 74 |
| | White | 1730 |
| | Unknown | 664 |
| Ethnicity | Hispanic | 495 |
| | Not Hispanic | 2612 |
| | Unknown | 107 |

Table 20: Subgroup stratified counts for the test split under INSPECT dataset

| | | AUROC ↑ | | | |
|---|---|---|---|---|---|
| Group | Concept | SwinUNETR $_{base}$ | SwinUNETR $_{base/MTL}$ | SwinUNETR $_{base/visit}$ | SwinUNETR $_{base/TTE}$ |
| Gender | Female | 0.666 (0.607, 0.722) | 0.648 (0.588, 0.705) | 0.690 (0.636, 0.739) | **0.828 (0.789, 0.866)** |
| | Male | 0.735 (0.670, 0.800) | 0.712 (0.637, 0.785) | 0.696 (0.634, 0.753) | **0.841 (0.797, 0.879)** |
| Age | 18-39 | 0.753 (0.591, 0.887) | 0.715 (0.553, 0.863) | 0.643 (0.512, 0.765) | **0.887 (0.808, 0.952)** |
| | 39-69 | 0.701 (0.639, 0.755) | 0.688 (0.626, 0.751) | 0.683 (0.628, 0.735) | **0.846 (0.809, 0.882)** |
| | 69-89 | 0.661 (0.580, 0.738) | 0.643 (0.558, 0.720) | 0.730 (0.672, 0.790) | **0.809 (0.752, 0.864)** |
| | >89 | 0.662 (0.435, 0.869) | 0.648 (0.434, 0.877) | **0.723 (0.445, 0.916)** | 0.647 (0.426, 0.847) |
| Race | Asian | 0.675 (0.547, 0.787) | 0.642 (0.522, 0.760) | 0.668 (0.562, 0.774) | **0.831 (0.732, 0.912)** |
| | Black | 0.533 (0.328, 0.738) | 0.512 (0.313, 0.707) | 0.660 (0.523, 0.785) | **0.805 (0.690, 0.905)** |
| | Native | 0.667 (0.040, 1.000) | 0.689 (0.040, 1.000) | 0.743 (0.333, 0.997) | **0.773 (0.555, 0.939)** |
| | White | 0.749 (0.694, 0.797) | 0.732 (0.673, 0.786) | 0.718 (0.665, 0.769) | **0.855 (0.816, 0.889)** |
| | Unknown | 0.649 (0.561, 0.733) | 0.636 (0.553, 0.726) | 0.670 (0.598, 0.740) | **0.778 (0.719, 0.830)** |
| Ethnicity | Hispanic | 0.645 (0.532, 0.754) | 0.646 (0.524, 0.758) | 0.685 (0.588, 0.778) | **0.785 (0.701, 0.866)** |
| | Not Hispanic | 0.701 (0.648, 0.751) | 0.681 (0.630, 0.730) | 0.699 (0.655, 0.741) | **0.842 (0.809, 0.873)** |
| | Unknown | 0.731 (0.563, 0.862) | 0.720 (0.566, 0.848) | 0.688 (0.572, 0.800) | **0.832 (0.744, 0.912)** |

Table 21: 1-Month Mortality prognosis breakdown by subgroups, reported as the mean AUROC of a test set bootstrap (n=1000) with 95% CI. **Bold** indicates the best performance across all models.

| | | AUROC ↑ | | | |
|---|---|---|---|---|---|
| Group | Concept | SwinUNETR $_{base}$ | SwinUNETR $_{base/MTL}$ | SwinUNETR $_{base/visit}$ | SwinUNETR $_{base/TTE}$ |
| Gender | Female | 0.661 (0.620, 0.702) | 0.676 (0.632, 0.717) | 0.699 (0.653, 0.743) | **0.813 (0.779, 0.842)** |
| | Male | 0.710 (0.657, 0.762) | 0.739 (0.688, 0.790) | 0.727 (0.683, 0.769) | **0.819 (0.779, 0.858)** |
| Age | 18-39 | 0.799 (0.727, 0.861) | 0.803 (0.729, 0.867) | 0.723 (0.634, 0.810) | **0.887 (0.820, 0.944)** |
| | 39-69 | 0.685 (0.641, 0.725) | 0.700 (0.653, 0.747) | 0.690 (0.645, 0.734) | **0.811 (0.775, 0.842)** |
| | 69-89 | 0.635 (0.571, 0.691) | 0.657 (0.596, 0.717) | 0.755 (0.693, 0.805) | **0.799 (0.752, 0.844)** |
| | >89 | 0.680 (0.502, 0.844) | **0.738 (0.562, 0.880)** | 0.687 (0.473, 0.870) | 0.732 (0.528, 0.883) |
| Race | Asian | 0.666 (0.583, 0.740) | 0.693 (0.608, 0.771) | 0.682 (0.591, 0.770) | **0.755 (0.667, 0.840)** |
| | Black | 0.573 (0.414, 0.728) | 0.615 (0.478, 0.758) | 0.666 (0.536, 0.793) | **0.818 (0.720, 0.905)** |
| | Native | 0.806 (0.500, 1.000) | 0.794 (0.488, 0.986) | 0.795 (0.586, 0.966) | **0.882 (0.710, 0.990)** |
| | White | 0.713 (0.674, 0.751) | 0.724 (0.684, 0.765) | 0.733 (0.694, 0.772) | **0.834 (0.802, 0.862)** |
| | Unknown | 0.664 (0.612, 0.713) | 0.680 (0.627, 0.728) | 0.707 (0.650, 0.761) | **0.779 (0.739, 0.818)** |
| Ethnicity | Hispanic | 0.651 (0.554, 0.743) | 0.673 (0.579, 0.762) | 0.679 (0.597, 0.756) | **0.767 (0.694, 0.833)** |
| | Not Hispanic | 0.688 (0.653, 0.724) | 0.709 (0.675, 0.743) | 0.724 (0.688, 0.759) | **0.820 (0.795, 0.845)** |
| | Unknown | 0.693 (0.624, 0.758) | 0.689 (0.625, 0.753) | 0.726 (0.647, 0.795) | **0.792 (0.741, 0.840)** |

Table 22: 6-Month Mortality prognosis breakdown by subgroups, reported as the mean AUROC of a test set bootstrap (n=1000) with 95% CI. **Bold** indicates the best performance across all models.

| | | AUROC ↑ | | | |
|---|---|---|---|---|---|
| Group | Concept | SwinUNETR $_{base}$ | SwinUNETR $_{base/MTL}$ | SwinUNETR $_{base/visit}$ | SwinUNETR $_{base/TTE}$ |
| Gender | Female | 0.655 (0.616, 0.694) | 0.669 (0.630, 0.706) | 0.660 (0.617, 0.702) | **0.796 (0.762, 0.825)** |
| | Male | 0.717 (0.669, 0.762) | 0.732 (0.685, 0.775) | 0.672 (0.626, 0.717) | **0.796 (0.761, 0.833)** |
| Age | 18-39 | 0.828 (0.760, 0.886) | 0.839 (0.777, 0.893) | 0.681 (0.589, 0.764) | **0.876 (0.805, 0.936)** |
| | 39-69 | 0.682 (0.636, 0.722) | 0.694 (0.653, 0.731) | 0.644 (0.598, 0.686) | **0.791 (0.759, 0.825)** |
| | 69-89 | 0.632 (0.573, 0.687) | 0.647 (0.589, 0.702) | 0.706 (0.649, 0.763) | **0.784 (0.738, 0.827)** |
| | >89 | 0.605 (0.434, 0.771) | 0.616 (0.429, 0.796) | 0.632 (0.440, 0.809) | **0.660 (0.481, 0.819)** |
| Race | Asian | 0.675 (0.604, 0.746) | 0.695 (0.624, 0.763) | 0.657 (0.567, 0.741) | **0.754 (0.684, 0.824)** |
| | Black | 0.624 (0.498, 0.746) | 0.645 (0.522, 0.766) | 0.620 (0.505, 0.730) | **0.781 (0.683, 0.866)** |
| | Native | 0.716 (0.451, 0.950) | 0.690 (0.394, 0.937) | 0.745 (0.487, 0.952) | **0.840 (0.650, 0.981)** |
| | White | 0.716 (0.675, 0.755) | 0.724 (0.687, 0.762) | 0.676 (0.632, 0.714) | **0.822 (0.793, 0.851)** |
| | Unknown | 0.657 (0.609, 0.703) | 0.673 (0.625, 0.719) | 0.672 (0.621, 0.725) | **0.750 (0.707, 0.788)** |
| Ethnicity | Hispanic | 0.643 (0.553, 0.728) | 0.660 (0.571, 0.746) | 0.631 (0.551, 0.701) | **0.708 (0.630, 0.780)** |
| | Not Hispanic | 0.689 (0.657, 0.721) | 0.702 (0.668, 0.736) | 0.675 (0.640, 0.714) | **0.808 (0.782, 0.835)** |
| | Unknown | 0.698 (0.638, 0.755) | 0.706 (0.647, 0.761) | 0.695 (0.621, 0.764) | **0.764 (0.713, 0.811)** |

Table 23: 12-Month Mortality prognosis breakdown by subgroups, reported as the mean AUROC of a test set bootstrap (n=1000) with 95% CI. **Bold** indicates the best performance across all models.

| | | AUROC ↑ | | | |
|---|---|---|---|---|---|
| Group | Concept | SwinUNETR $_{base}$ | SwinUNETR $_{base/MTL}$ | SwinUNETR $_{base/visit}$ | SwinUNETR $_{base/TTE}$ |
| Gender | Female | 0.539 (0.459, 0.623) | 0.534 (0.457, 0.614) | 0.551 (0.477, 0.630) | **0.599 (0.522, 0.671)** |
| | Male | 0.498 (0.391, 0.607) | 0.492 (0.386, 0.600) | 0.532 (0.448, 0.613) | **0.587 (0.499, 0.675)** |
| Age | 18-39 | 0.452 (0.280, 0.634) | 0.438 (0.260, 0.613) | 0.598 (0.456, 0.737) | **0.603 (0.423, 0.760)** |
| | 39-69 | 0.458 (0.373, 0.546) | 0.457 (0.374, 0.546) | 0.523 (0.446, 0.597) | **0.558 (0.482, 0.628)** |
| | 69-89 | 0.687 (0.573, 0.794) | **0.689 (0.581, 0.787)** | 0.492 (0.370, 0.608) | 0.676 (0.547, 0.787) |
| | >89 | 0.459 (0.013, 0.907) | 0.432 (0.013, 0.878) | **0.772 (0.585, 0.915)** | 0.363 (0.200, 0.533) |
| Race | Asian | 0.594 (0.442, 0.742) | 0.603 (0.428, 0.748) | **0.674 (0.485, 0.854)** | 0.619 (0.460, 0.763) |
| | Black | 0.374 (0.025, 0.848) | 0.397 (0.027, 0.865) | **0.523 (0.338, 0.775)** | 0.524 (0.201, 0.945) |
| | Native | 0.860 (0.760, 0.940) | 0.880 (0.780, 0.960) | 0.287 (0.146, 0.447) | **0.941 (0.875, 1.000)** |
| | White | 0.490 (0.411, 0.573) | 0.478 (0.395, 0.558) | 0.521 (0.449, 0.591) | **0.614 (0.540, 0.686)** |
| | Unknown | 0.491 (0.408, 0.571) | 0.489 (0.410, 0.575) | **0.613 (0.525, 0.700)** | 0.525 (0.444, 0.604) |
| Ethnicity | Hispanic | **0.670 (0.479, 0.836)** | 0.662 (0.468, 0.842) | 0.515 (0.406, 0.626) | 0.587 (0.400, 0.774) |
| | Not Hispanic | 0.515 (0.439, 0.586) | 0.513 (0.442, 0.587) | 0.542 (0.475, 0.610) | **0.620 (0.557, 0.680)** |
| | Unknown | 0.433 (0.347, 0.522) | 0.426 (0.340, 0.511) | **0.666 (0.563, 0.761)** | 0.480 (0.380, 0.576) |

Table 24: 1-Month Readmission prognosis breakdown by subgroups, reported as the mean AUROC of a test set bootstrap (n=1000) with 95% CI. **Bold** indicates the best performance across all models.

| Group | Concept | AUROC ↑ | | | |
| | | SwinUNETR $_{\text{base}}$ | SwinUNETR $_{\text{base/MTL}}$ | SwinUNETR $_{\text{base/visit}}$ | SwinUNETR $_{\text{base/TTE}}$ |
|---|---|---|---|---|---|
| Gender | Female | 0.571 (0.522, 0.619) | 0.577 (0.528, 0.626) | 0.488 (0.436, 0.541) | **0.607 (0.558, 0.654)** |
| | Male | 0.525 (0.466, 0.585) | 0.526 (0.470, 0.583) | **0.608 (0.541, 0.670)** | 0.607 (0.556, 0.660) |
| Age | 18-39 | 0.533 (0.428, 0.630) | 0.542 (0.440, 0.636) | 0.541 (0.434, 0.654) | **0.660 (0.569, 0.753)** |
| | 39-69 | 0.516 (0.467, 0.564) | 0.519 (0.472, 0.566) | 0.519 (0.461, 0.573) | **0.590 (0.544, 0.634)** |
| | 69-89 | 0.616 (0.542, 0.691) | **0.624 (0.549, 0.694)** | 0.537 (0.456, 0.616) | 0.610 (0.535, 0.689) |
| | >89 | 0.604 (0.000, 0.984) | 0.616 (0.000, 0.984) | **0.792 (0.601, 0.956)** | 0.553 (0.161, 0.921) |
| Race | Asian | 0.548 (0.456, 0.641) | 0.556 (0.459, 0.648) | 0.549 (0.401, 0.693) | **0.591 (0.497, 0.678)** |
| | Black | 0.510 (0.305, 0.706) | 0.509 (0.305, 0.698) | 0.525 (0.365, 0.697) | **0.557 (0.382, 0.734)** |
| | Native | 0.841 (0.612, 0.989) | **0.850 (0.659, 0.985)** | 0.598 (0.431, 0.774) | 0.835 (0.588, 0.977) |
| | White | 0.554 (0.504, 0.604) | 0.561 (0.511, 0.610) | 0.529 (0.479, 0.580) | **0.626 (0.575, 0.675)** |
| | Unknown | 0.504 (0.451, 0.559) | 0.502 (0.451, 0.553) | 0.591 (0.522, 0.652) | **0.594 (0.544, 0.642)** |
| Ethnicity | Hispanic | **0.610 (0.516, 0.704)** | 0.606 (0.510, 0.695) | 0.594 (0.503, 0.685) | 0.556 (0.455, 0.648) |
| | Not Hispanic | 0.553 (0.507, 0.596) | 0.560 (0.519, 0.601) | 0.528 (0.475, 0.574) | **0.620 (0.581, 0.661)** |
| | Unknown | 0.469 (0.401, 0.532) | 0.472 (0.413, 0.534) | 0.600 (0.511, 0.687) | **0.605 (0.541, 0.665)** |

Table 25: 6-Month Readmission prognosis breakdown by subgroups, reported as the mean AUROC of a test set bootstrap (n=1000) with 95% CI. **Bold** indicates the best performance across all models.

| Group | Concept | AUROC ↑ | | | |
| | | SwinUNETR $_{\text{base}}$ | SwinUNETR $_{\text{base/MTL}}$ | SwinUNETR $_{\text{base/visit}}$ | SwinUNETR $_{\text{base/TTE}}$ |
|---|---|---|---|---|---|
| Gender | Female | 0.588 (0.539, 0.629) | 0.565 (0.520, 0.609) | 0.492 (0.444, 0.538) | **0.607 (0.562, 0.650)** |
| | Male | 0.563 (0.512, 0.612) | 0.547 (0.497, 0.598) | 0.555 (0.501, 0.610) | **0.601 (0.552, 0.650)** |
| Age | 18-39 | 0.557 (0.467, 0.645) | 0.554 (0.462, 0.639) | 0.496 (0.402, 0.581) | **0.653 (0.573, 0.730)** |
| | 39-69 | 0.562 (0.519, 0.605) | 0.530 (0.484, 0.571) | 0.509 (0.461, 0.557) | **0.596 (0.554, 0.640)** |
| | 69-89 | **0.611 (0.544, 0.673)** | 0.600 (0.529, 0.664) | 0.529 (0.464, 0.596) | 0.592 (0.523, 0.659) |
| | >89 | 0.589 (0.019, 0.913) | 0.675 (0.180, 0.981) | **0.678 (0.466, 0.875)** | 0.564 (0.130, 0.855) |
| Race | Asian | 0.614 (0.524, 0.694) | 0.587 (0.497, 0.679) | 0.473 (0.383, 0.561) | **0.635 (0.555, 0.716)** |
| | Black | 0.494 (0.336, 0.680) | 0.486 (0.325, 0.642) | 0.488 (0.344, 0.635) | **0.634 (0.472, 0.787)** |
| | Native | **0.901 (0.763, 0.987)** | 0.843 (0.679, 0.976) | 0.631 (0.462, 0.792) | 0.861 (0.658, 1.000) |
| | White | 0.589 (0.542, 0.632) | 0.564 (0.519, 0.607) | 0.516 (0.472, 0.562) | **0.626 (0.585, 0.668)** |
| | Unknown | 0.523 (0.481, 0.569) | 0.515 (0.470, 0.560) | **0.566 (0.510, 0.621)** | 0.560 (0.512, 0.607) |
| Ethnicity | Hispanic | 0.522 (0.440, 0.604) | 0.534 (0.447, 0.622) | **0.602 (0.510, 0.689)** | 0.479 (0.401, 0.557) |
| | Not Hispanic | 0.597 (0.560, 0.634) | 0.570 (0.534, 0.608) | 0.506 (0.464, 0.545) | **0.635 (0.601, 0.669)** |
| | Unknown | 0.509 (0.454, 0.565) | 0.501 (0.437, 0.557) | **0.554 (0.472, 0.631)** | 0.589 (0.529, 0.644) |

Table 26: 12-Month Readmission prognosis breakdown by subgroups, reported as the mean AUROC of a test set bootstrap (n=1000) with 95% CI. **Bold** indicates the best performance across all models.

| Group | Concept | AUROC ↑ | | | |
| | | SwinUNETR $_{\text{base}}$ | SwinUNETR $_{\text{base/MTL}}$ | SwinUNETR $_{\text{base/visit}}$ | SwinUNETR $_{\text{base/TTE}}$ |
|---|---|---|---|---|---|
| Gender | Female | 0.597 (0.547, 0.645) | 0.607 (0.556, 0.655) | 0.559 (0.515, 0.602) | **0.665 (0.618, 0.710)** |
| | Male | 0.563 (0.504, 0.622) | 0.572 (0.512, 0.631) | 0.542 (0.496, 0.592) | **0.673 (0.625, 0.721)** |
| Age | 18-39 | 0.543 (0.379, 0.709) | 0.553 (0.386, 0.703) | 0.515 (0.431, 0.594) | **0.758 (0.650, 0.862)** |
| | 39-69 | 0.606 (0.552, 0.660) | 0.615 (0.562, 0.669) | 0.552 (0.506, 0.596) | **0.661 (0.614, 0.705)** |
| | 69-89 | 0.547 (0.486, 0.609) | 0.556 (0.497, 0.616) | 0.579 (0.516, 0.639) | **0.644 (0.584, 0.705)** |
| | >89 | 0.539 (0.361, 0.731) | **0.548 (0.361, 0.728)** | 0.534 (0.404, 0.669) | 0.464 (0.294, 0.631) |
| Race | Asian | 0.613 (0.521, 0.707) | 0.620 (0.523, 0.714) | 0.547 (0.470, 0.627) | **0.667 (0.576, 0.750)** |
| | Black | 0.706 (0.532, 0.850) | 0.727 (0.562, 0.876) | 0.373 (0.249, 0.501) | **0.767 (0.644, 0.872)** |
| | Native | 0.378 (0.074, 0.711) | 0.326 (0.046, 0.644) | 0.218 (0.075, 0.369) | **0.583 (0.312, 0.833)** |
| | White | 0.579 (0.529, 0.630) | 0.593 (0.543, 0.643) | 0.580 (0.539, 0.625) | **0.687 (0.643, 0.730)** |
| | Unknown | 0.603 (0.558, 0.647) | 0.609 (0.566, 0.655) | 0.581 (0.532, 0.633) | **0.641 (0.595, 0.686)** |
| Ethnicity | Hispanic | 0.602 (0.511, 0.687) | 0.615 (0.521, 0.703) | 0.500 (0.414, 0.584) | **0.641 (0.550, 0.722)** |
| | Not Hispanic | 0.584 (0.538, 0.626) | 0.589 (0.547, 0.632) | 0.562 (0.526, 0.598) | **0.679 (0.643, 0.715)** |
| | Unknown | 0.618 (0.566, 0.672) | 0.626 (0.570, 0.677) | 0.597 (0.526, 0.662) | **0.646 (0.594, 0.696)** |

Table 27: 12-Month Pulmonary Hypertension prognosis breakdown by subgroups, reported as the mean AUROC of a test set bootstrap (n=1000) with 95% CI. **Bold** indicates the best performance across all models.

