# OpenReview forum: "Time-to-Event Pretraining for 3D Medical Imaging"
_ICLR.cc/2025/Conference — ICLR 2025 Poster_

### Official Review · Reviewer_iczn · 2024-10-20

**Soundness:** 3
**Presentation:** 4
**Contribution:** 3
**Rating:** 8
**Confidence:** 4

**Summary:**

*Edit: Score increased from 6 to 8 during discussion period.*

This paper presents a self-supervised learning (SSL) method for 3D medical imaging data that leverages electronic health records (EHR) to provide extra sources of supervision via time-to-event modeling. The proposed method, future-guided pretraining, performs time-to-event (TTE) survival modeling of various medical events in the longitudinal EHR associated with each 3D scan. The authors show that future-guided pretraining consistently improves downstream TTE modeling and prognostic classification tasks – also improving data efficiency – without degrading standard diagnostic classification performance.

**Strengths:**

- The presentation quality is very high. Care has been taken to logically organize the paper, clearly articulate key points, and straightforwardly present results with concise figures and tables.
- The core idea is creative, making use of the wealth of longitudinal EHR data associated with each 3D volume for pretraining.
- Discussion or related work and background is particularly strong.
- Experiments are sufficiently thorough and easy to interpret – results are convincing.

**Weaknesses:**

- The actual description of the TTE pretraining approach is brief (lines 184-191) and somewhat unclear. I would advise the authors to flesh out this section. See specific questions below.
- A description or list of the 8,192 EHR pretraining tasks is never provided. I’m aware there may not be a convenient place to list this many items, but a general description of categories of events or a few illustrative examples would be helpful. Without this information, it’s impossible to assess whether, e.g., one the TTE pretraining tasks is *also* used as a downstream TTE modeling task. In this case, there may be concerns of “label leakage”.

I’m happy to increase my score once these issues are addressed – this is an otherwise strong submission.

**Questions:**

- What exactly does it mean that Steinberg et al.’s method was used to “[sample tasks to maximize entropy given the frequency distribution of medical codes populating the DAG”? I feel that a basic plain-language description of the motivation for this procedure is needed first: why is this method being applied at all? Are there way more than 8k events and the goal is to settle on a subset of 8k “meaningful”/common ones for pretraining? I don’t understand the motivation.
- Unless I am misunderstanding, this is the only description of the TTE pretraining procedure and labels used: “We define our TTE task labels by predicting the time until the next occurrence of a medical code.” The previous Section 3 described deep survival modeling in the abstract, so I expected Section 4 to more concretely describe how TTE pretraining works. Is this a “competing risks” approach, where multiple events are being modeled simultaneously (in “multi-label” fashion)?
- What are the 8,192 EHR tasks/events? I’m aware it would be cumbersome or impossible to list and define them all, but any reasonable attempt to convey information about them would be useful. What kinds of “events” are they? What are some examples?
- Related to the above point, are the downstream labels *also* present in the set of TTE pretraining tasks? If so, isn’t there concern of “label leakage”, where the model has been pretrained on label information present in the downstream training dataset? Please clarify this.

**Minor comments/questions:**
- Line 13: Maybe “build” instead of “capture” since you use this word in the next sentence.
- In-text citation style seems off – should be parenthetical (\pcite{}) in most cases when used at end of sentence/clause: “Sox et al. (2024)” -> “(Sox et al., 2024)”
- Change “e.g.” -> “e.g.,” throughout
- Would include more recent references [1,2] when discussing deep prognosis models on longitudinal medical imaging (first paragraph of Section 2)
- “i.e. 8192” -> “i.e., 8.192”
- “Our approach improves training data efficiency, increasing training labels by an average of 3x over labels assigned to patients based on their current EHR visit.” This is a bit unusual to highlight as a main contribution – I don’t think readers will understand what “increasing training labels” means without having read the entire paper (nor why this impact data efficiency). Perhaps clarify language here to indicate that your approach provides 3x as many sources of supervision during SSL + that this is what provides data efficiency benefits.
- “Pretraining task labels as assigned per-CT scan and vary in density based on pretraining approach, see Figure 2.” Perhaps “as assigned” is meant to be “are assigned”? Also change “, see Figure 2” -> “(Figure 2)”.
- Be consistent with “c-statistic” vs. “C-statistic”

**References**
[1] Holste, Gregory, et al. "Harnessing the power of longitudinal medical imaging for eye disease prognosis using Transformer-based sequence modeling." NPJ Digital Medicine 7.1 (2024): 216.
[2] Sriram, Anuroop, et al. "Covid-19 prognosis via self-supervised representation learning and multi-image prediction." arXiv preprint arXiv:2101.04909 (2021).

---

> ### Author Response · Authors · 2024-11-21
> **Response to reviewer iczn**
>
> We thank reviewer iczn for the thoughtful comments. We now address them individually:
>
> > The actual description of the TTE pretraining approach is brief (lines 184-191) and somewhat unclear. I would advise the authors to flesh out this section. See specific questions below.
>
> We thank the reviewer for the suggestions. We have added Appendix L to illustrate the details of piecewise exponential loss function, which can be seen as a drop-in replacement of common loss function, such as cross-entropy, when each loss is calculated and aggregated on each image given a pre-defined set (i.e. 8,192) of TTE tasks.
>
> > A description or list of the 8,192 EHR pretraining tasks is never provided. I’m aware there may not be a convenient place to list this many items, but a general description of categories of events or a few illustrative examples would be helpful. Without this information, it’s impossible to assess whether, e.g., one the TTE pretraining tasks is also used as a downstream TTE modeling task. In this case, there may be concerns of “label leakage”.
>
> > What are the 8,192 EHR tasks/events? I’m aware it would be cumbersome or impossible to list and define them all, but any reasonable attempt to convey information about them would be useful. What kinds of “events” are they? What are some examples?
>
> > Related to the above point, are the downstream labels also present in the set of TTE pretraining tasks? If so, isn’t there concern of “label leakage”, where the model has been pretrained on label information present in the downstream training dataset? Please clarify this.
>
> We added Appendix O for the examples of pretraining tasks, by stratifying them into quintiles based on frequency on the cohort, and listed top 3 codes along with their descriptions in each quintile (e.g. LOINC codes for medical measurements, RxNorm codes for clinical drugs, ICD codes for disease classification). We additionally added Appendix N to illustrate the the event time distribution for the 8,192 pretraining tasks in terms of event (each event is one pretraining task) and occurrence (each event may have many occurrences across all the cohort). We had made sure the 8,192 pretraining tasks don’t contain any downstream evaluation labels in our experiment. We make sure the language in the manuscript aligns with it.
>
> > What exactly does it mean that Steinberg et al.’s method was used to “[sample tasks to maximize entropy given the frequency distribution of medical codes populating the DAG”? I feel that a basic plain-language description of the motivation for this procedure is needed first: why is this method being applied at all? Are there way more than 8k events and the goal is to settle on a subset of 8k “meaningful”/common ones for pretraining? I don’t understand the motivation.
>
> We apologize for the lack of clarity and present the plain-language motivation here. The most common way of selecting EHR features is usually by frequency based ranking [1], which is effectively a ranking by Shannon entropy (i.e. taking the most frequently used codes in the dataset). However, due to the nature of medical codes materialized in ontology (e.g. ICD, SNOMED), each code’s ranking should reflect the conditional probability of presence or absence of its parent (or ancestor) codes. The intuition is that a patient must have higher level codes if they have a lower level code so there’s a diminishing return for selecting lower level codes when their parent codes are selected. Our process effectively makes the setting as a vertex cover problem in directed acyclic graph (DAG), to cover as many ‘informative’ vertices as possible through increasing Shannon entropy. We have in total 4.3 million unique medical codes in our cohort in the post-scan period for all patients thus selecting all codes would be an impractical solution. We therefore rank the codes per above given a task budget for pretraining. We added explanations in Section 4.

---

> ### Author Response · Authors · 2024-11-21
> **Response to reviewer iczn (cont'd)**
>
> > Unless I am misunderstanding, this is the only description of the TTE pretraining procedure and labels used: “We define our TTE task labels by predicting the time until the next occurrence of a medical code.” The previous Section 3 described deep survival modeling in the abstract, so I expected Section 4 to more concretely describe how TTE pretraining works. Is this a “competing risks” approach, where multiple events are being modeled simultaneously (in “multi-label” fashion)?
>
> We apologize for the lack of detail. We have rewritten Section 4 and Appendix L to expand upon and clarify our pretraining regime. In summary, each TTE task (n=8,192) is modeled independently where death is the only competing risk. The survival loss is optimized for the first occurrence of an event, which is a common practice in time-to-event modeling [2].
>
> [1] Scheurwegs, Elyne, et al. "Selecting relevant features from the electronic health record for clinical code prediction." Journal of biomedical informatics 74 (2017): 92-103.
>
> [2] Liu, Jianfang, et al. "An integrated TCGA pan-cancer clinical data resource to drive high-quality survival outcome analytics." Cell 173.2 (2018): 400-416.

---

> ### Comment · Reviewer_iczn · 2024-11-25
>
> I have read the authors' rebuttal and appreciate the attention to detail in incorporating feedback. My mostly minor concerns have been addressed, and I am happy to raise my score to an 8.

---

### Official Review · Reviewer_Wo2C · 2024-11-03

**Soundness:** 2
**Presentation:** 3
**Contribution:** 2
**Rating:** 5
**Confidence:** 4

**Summary:**

The authors proposed to utilize the time-to-event information in EHR that paired with the imaging data as a form of supervision for the pre-training purpose. A public dataset with both 3D images and EHR notes is employed for the pre-training and downstream applications. Another dataset without the time events is also used for the evaluation of model adaptation. The manuscript is easy to follow. However, it also suffers from several critical flaws, which are detailed below.

**Strengths:**

- Propose utilizing the time events as pre-training tasks specially designed for prognosis tasks in downstream applications.
- The manuscript is overall easy to follow

**Weaknesses:**

- The proposed method is limited in generalization since it will require longitudinal time-to-event EHR data as the supervision for the pre-training. In comparison to the common self-supervised pre-training, the proposed methods are harder to scale up.

- There is no comparison evaluation between the proposed method and prior methods in model pre-training. Only the results of the proposed method with different model architectures are reported. It will be difficult to appreciate the benefits of the proposed method.

- The selected model architecture also raises questions since there are many popular model networks in medical imaging, e.g., 3D-UNet, ViT, etc. It will be helpful to see their performance compared to the vanilla ResNet.

- Baselines without the pre-training process should also be reported.

- The current setting utilizes public data for both pre-training and downstream applications. Having a separate evaluation dataset of a prognosis task will be helpful.

- The proposed method is limited in technical innovation, though utilizing the time-to-event data as a form of supervision is relatively new in the pre-training. Mostly existing techniques are adopted for the pre-training.

**Questions:**

See above

---

> ### Author Response · Authors · 2024-11-21
> **Response to reviewer Wo2C**
>
> We thank reviewer Wo2C for the thoughtful comments. We now address them individually:
>
> > The proposed method is limited in generalization since it will require longitudinal time-to-event EHR data as the supervision for the pre-training. In comparison to the common self-supervised pre-training, the proposed methods are harder to scale up.
>
> While we appreciate the criticism, we should point out that the assumption here is not correct. EHR is universal in all hospitals (30% of world data volume is produced in the healthcare system [5]). So in practice, EHR data is always available when CT is captured in any healthcare setting realistically. In our argument, pairing with readily available EHR for medical imaging modeling, we can scale up more easily whenever new medical imaging data is available (since the EHR should always be present, tens of thousands of orders of magnitude larger than the scale of imaging data [4]).
>
> > There is no comparison evaluation between the proposed method and prior methods in model pre-training. Only the results of the proposed method with different model architectures are reported. It will be difficult to appreciate the benefits of the proposed method.
>
> We have expanded upon Appendix D for comparison with Merlin [1] which is a suite of 3D imaging foundation models but without future guided pretraining (i.e. local diagnostic codes and radiology reports). The performance of their classification model (they employed ResNet as backbone for classification task) has shown to be suboptimal, where they are worse on 9/24 tasks and statistically indistinguishable in 10/24 tasks. We hypothesize that it can be (1) their local supervision signal in nature can’t predict well long term outcomes (2) their abdominal CTs are out-of-distribution from our focus for pulmonary disease.
> Furthermore, each of our model’s \base variant is a ‘pretrained’ model, either from medical dataset or generic dataset (see Table 10). Thus in our Discussion and Conclusion section, we highlight the comparison between models \TTE variant and the \base variant and the performance gap, to show our pretraining method is better for prognosis tasks than baseline pretraining techniques (either MAE style or supervised style).
>
> > The selected model architecture also raises questions since there are many popular model networks in medical imaging, e.g., 3D-UNet, ViT, etc. It will be helpful to see their performance compared to the vanilla ResNet.
>
> We thank the reviewer for the suggestion. However, as we have pointed out, our approach is in the category of pretraining technique which should be architecture agnostic, and we have chosen a representative set of model architectures (ViT-based with larger params vs. CNN-based, more lightweight) : (1) SwinUNETR: a strong medical imaging model, with pretraining weights as in-domain and dataset is larger than other common medical imaging models [2]. (2) ResNet: it’s commonly used in other 3D foundation models [1] and we employed a similar weight inflation technique, thus not a vanilla ResNet. (3) DenseNet: a very versatile and lightweight imaging model, that has shown to be faster in convergence [3]. Plus the model parameter is relatively smaller to accommodate restricted compute environments compared to transformer-based models. (For our case, SwinUNETR has > 80 million parameters, ResNet has 60 million and DenseNet has 11 million). We aim to use these 3 different architectures to cover a wide range of image backbones. We are, on the other hand, hoping to expand to wider architectures for ablations in future work.
>
> > Baselines without the pre-training process should also be reported.
>
> We agree on the importance of ablating pretraining. Our original paper does report ablation baselines without our TTE pretraining method (in Table 2, 3, 4, all the ‘base’ models are either pretrained from in-domain MRI data or generic ImageNet). We have updated the manuscript's writing to make this clearer to the reviewers. We have also expanded Appendix K for all the model architectures with random initialization as comparison. On average the random init version provides nearly no predictive power (i.e. AUROC is 0.5 to 0.56 Harrell’s C-index is 0.5 to 0.62, and poor calibration, i.e. up to 0.84 IBS).
>
> > The current setting utilizes public data for both pre-training and downstream applications. Having a separate evaluation dataset of a prognosis task will be helpful.
>
> We appreciate the rigor for external validation of our models. To evaluate our method we need chest CT scans linked to longitudinal outcomes. To our knowledge, there are no such datasets other than INSPECT [4]. The need for more prognostic evaluation tasks is what motivated us to add 8 additional prognostic EHR tasks defined on INSPECT. Our hope is that with this work’s results, we can shed light on the need of larger scale medical imaging datasets along with EHR modality, and enthuse the community of joint efforts to produce them thereof.

---

> ### Author Response · Authors · 2024-11-21
> **Response to reviewer Wo2C (cont'd)**
>
> > The proposed method is limited in technical innovation, though utilizing the time-to-event data as a form of supervision is relatively new in the pre-training. Mostly existing techniques are adopted for the pre-training.
>
> We appreciate the criticism for technical innovation. However, as we stated in the paper, the novelty lies in applying TTE pretraining at scale to 3D imaging data, which has not been done before. We have revised the corresponding Introduction section to elucidate the state of affairs for the medical imaging domain where the need for prognostic information is not met by current methods under missing context problem (see new Figure 1, where most methods only focus on local EHR/textual pairings for pretraining and missing the longitudinal aspect) and thus we propose to lay the groundwork for methods that can learn information from the future as supervision signals. We further changed the paper title to avoid overclaiming the contribution and pinpointed time-to-event as the narrowly scoped methodology contribution.
>
> [1] Blankemeier, Louis, et al. "Merlin: A Vision Language Foundation Model for 3D Computed Tomography." arXiv preprint arXiv:2406.06512 (2024).
>
> [2] Wasserthal, Jakob, et al. "TotalSegmentator: robust segmentation of 104 anatomic structures in CT images." Radiology: Artificial Intelligence 5.5 (2023).
>
> [3] Zhou, Tao, et al. "Dense convolutional network and its application in medical image analysis." BioMed Research International 2022.1 (2022): 2384830.
>
> [4] Huang, Shih-Cheng, et al. "INSPECT: a multimodal dataset for pulmonary embolism diagnosis and prognosis." Proceedings of the 37th International Conference on Neural Information Processing Systems. 2023.
>
> [5] Thomason, Jane. "Big tech, big data and the new world of digital health." Global Health Journal 5.4 (2021): 165-168.

---

> ### Comment · Reviewer_Wo2C · 2024-11-22
>
> I thank the authors for the detailed responses to my previous comments, which addressed some of my concerns. However, one of the critical concerns about the lack of comparison to previous pre-training models (especially those with self-supervised learning) remains. Merlin is not a foundation model, and it is also not trained on 3D medical data (2.5D, actually). Without such proper comparison, it is hard to appreciate the benefit of utilizing the proposed longitudinal information as the additional supervision.
>
> BTW, the illustrated CT images are not properly displayed, i.e., wrong position and window.

---

> ### Author Response · Authors · 2024-11-22
>
> > Comment: I thank the authors for the detailed responses to my previous comments, which addressed some of my concerns. However, one of the critical concerns about the lack of comparison to previous pre-training models (especially those with self-supervised learning) remains. Merlin is not a foundation model, and it is also not trained on 3D medical data (2.5D, actually). Without such proper comparison, it is hard to appreciate the benefit of utilizing the proposed longitudinal information as the additional supervision.
>
> Thank you for the fast response and the ability to clarify our contributions.
>
> > lack of comparison to previous pre-training models
>
> A core challenge in this space is the lack of existing native 3D pretrained medical foundation models to enable this type comparison. Merlin and SwinUNETR (both included in this work) were the largest pretrained 3D models available at time of submission. We believe these models represent the current state-of-the-art baselines for evaluating large-scale MAE (image-only) and contrastive (language-image) self-supervised learning approaches in 3D CT data.
>
> We welcome specific suggestions for additional pretrained 3D medical foundation models to include in this work and feedback on the experimental gaps their inclusion would address.
>
> > Merlin is not a foundation model, and it is also not trained on 3D medical data (2.5D, actually).
>
> Merlin [1] utilizes a true 3D architecture and is a foundation model pretrained on 3D abdominal CT scans. Their work explores multiple native 3D architectures (3D Swin Transformer, ConvNeXt, ResNet) and multiple downstream adapted tasks (segmentation, disease classification, prognosis, cross-modal retrieval, report generation, etc., thus a foundation model) under different sampling assumptions, e.g., zero-shot. They do not explore 2.5D architectures. We used the best performing variant of Merlin in our work, i.e. ResNet-152.
>
> > Without such proper comparison, it is hard to appreciate the benefit of utilizing the proposed longitudinal information as the additional supervision.
>
> Our experiments include a representative set of state-of-the-art (SOTA) baselines, reflect different SSL pretraining approaches for 3D CT data (Merlin, SwinUNETR) and 3D architectures. We believe the significant performance gains observed in these SOTA models following TTE pretraining strongly support the effectiveness of our approach.
>
> > BTW, the illustrated CT images are not properly displayed, i.e., wrong position and window.
>
> We have entirely redesigned and updated new figures per our general response. Please refer to new Figure 2 where the input CT is illustrated as a generic cube as we want to avoid visual confusion such as this.
>
>
> [1] Blankemeier, Louis, Joseph Paul Cohen, Ashwin Kumar, Dave Van Veen, Syed Jamal Safdar Gardezi, Magdalini Paschali, Zhihong Chen et al. "Merlin: A Vision Language Foundation Model for 3D Computed Tomography." arXiv preprint arXiv:2406.06512 (2024).

---

> > ### Comment · Reviewer_Wo2C · 2024-11-22
> >
> > 1. The following text is from the Merlin paper:
> > "Model Architecture: Merlin uses an inflated 3D (I3D) ResNet152 for the image encoder. Inflation refers to
> > reusing 2D pre-trained model weights and copying those weights across the 3rd dimension of the 3D convolutional
> > kernels [56]. "
> > That‘s why I call it 2.5D, and the block-based token in the transformer pipeline further suppressed the 3D spatial information.
> >
> > 2.  I just realized the base setting is the one using the vanilla pre-training model.
> >
> > 3. The images I mentioned are those in Figure 19. It is not a professional way of showing CT images, i.e., rotated axial images with improper windowing.
> >
> > Based on the first two clarified points, I would raise my rating to 5.

---

> ### Author Response · Authors · 2024-11-23
>
> > The following text is from the Merlin paper: "Model Architecture: Merlin uses an inflated 3D (I3D) ResNet152 for the image encoder. Inflation refers to reusing 2D pre-trained model weights and copying those weights across the 3rd dimension of the 3D convolutional kernels [56]. " That‘s why I call it 2.5D, and the block-based token in the transformer pipeline further suppressed the 3D spatial information.
>
> Thank you for these clarifications. Our manuscript should be clearer on differentiating **weight initialization** from **model architecture**.
>
> Merlin's ResNet parameters were initialized using 2D weight inflation, but the architecture is fully 3D, using 3D convolutions in each Res-block. This differs from 2.5D architectures [1]. The initialized ResNet architecture was then pretrained via contrastive SSL using 3D CT data, meaning it explicitly learns 3D relationships. Initializing from 2D inflated weights (vs random initialization) is a strategy that can lead to better performance when pretraining native 3D architectures, per the Merlin paper.
>
> For SwinUNETR, the architecture is fully 3D and hierarchical. The input volume is divided into voxel patches of size 2 with a channel number of 2, forming  2×2×2×2 tensors that are flattened into input token vectors. The Swin Transformer backbone explicitly learns 3D relationships across voxel patches via Swin Transformer Blocks, utilizing shifted window attention to capture interactions between neighboring patches [2,3,4]. The SwinUNETR model weights were randomly initialized and pre-trained from scratch using MAE on various medical 3D modalities such as CT scans and MRIs (i.e., no weight inflation was used).
>
> We apologize for the ambiguity in the manuscript on these important 3D architecture and weight initialization details, so we have updated the architecture section of the manuscript to improve clarity.
>
> > The images I mentioned are those in Figure 19. It is not a professional way of showing CT images, i.e., rotated axial images with improper windowing.
>
> Thank you for the feedback. The intention of this figure was to highlight gradient-based saliency maps. We acknowledge the challenges in representing 3D data in a 2D form that captures both the underlying CT scan and saliency maps. If the reviewer can point out specific visualization examples they feel are more effective or correct, we will be happy to update the figure.
>
> > Based on the first two clarified points, I would raise my rating to 5.
>
> Thank you for recognizing our efforts and helping us to improve the clarity and quality of our paper.
>
>
> Reference:
>
> [1] Hung, Alex Ling Yu, et al. "CSAM: A 2.5 D Cross-Slice Attention Module for Anisotropic Volumetric Medical Image Segmentation." Proceedings of the IEEE/CVF Winter Conference on Applications of Computer Vision. 2024.
>
> [2] Tang, Yucheng, et al. "Self-supervised pre-training of swin transformers for 3d medical image analysis." Proceedings of the IEEE/CVF conference on computer vision and pattern recognition. 2022.
>
> [3] Valanarasu, Jeya Maria Jose, et al. "Disruptive Autoencoders: Leveraging Low-level features for 3D Medical Image Pre-training." arXiv preprint arXiv:2307.16896 (2023).
>
> [4] He, Yufan, et al. "Swinunetr-v2: Stronger swin transformers with stagewise convolutions for 3d medical image segmentation." International Conference on Medical Image Computing and Computer-Assisted Intervention. Cham: Springer Nature Switzerland, 2023.

---

> ### Author Response · Authors · 2024-11-26
>
> Hi reviewer Wo2C,
>
> We thank you for your constructive criticism and timely follow-up for our responses throughout the rebuttal period. We are wondering if our newest comments address your concerns regarding:
>
> - 2D vs. 3D for **model architecture** and **weight initialization**
> - Further improvement on our visualization
>
> The discussion period is coming to an end and we hope the new manuscript brings more clarity to the comments. If not, what other aspect we can improve on?
>
> Regards,
>
> The authors

---

### Official Review · Reviewer_rQZZ · 2024-11-04

**Soundness:** 3
**Presentation:** 3
**Contribution:** 3
**Rating:** 6
**Confidence:** 4

**Summary:**

The paper introduces a future-guided pretraining approach using time-to-event supervision to enhance the prognostic capabilities of 3D medical imaging models. By incorporating longitudinal EHR data into the pretraining process and predicting time-until-event, the model outperforms traditional methods across multiple standard tasks, as demonstrated by thorough experiments.

**Strengths:**

1. Innovative Approach: The method creatively leverages EHR data following a medical scan to assist model pretraining, demonstrating better performance compared to imaging-only pretraining.
2. Comprehensive Evaluation: Extensive comparisons across multiple tasks validate the robustness and efficiency of the TTE-based approach across different architectures.

**Weaknesses:**

1. Dependence on Large EHR Datasets: This approach relies on extensive, high-quality EHR data, which many medical datasets do not include.
2. Limited Modality Scope: Tested only on CT images; broader modality testing could validate versatility across imaging types.
3. Interpretability: The TTE pretraining’s impact on specific pixel-level biomarkers is less clear; additional analysis on feature attribution could help.

**Questions:**

1. Why start from 3D image scans instead of 2D medical images? Is this due to the dataset choice, or has similar work already been done on 2D data?
2. How does the choice of time segmentation for EHR data affect model results during pretraining? Specifically, my understanding is that the model predicts the probability of a patient experiencing a certain event at intervals like 1, 2, or 3 years post-scan. How does the granularity of these time segments impact the performance of the pretrained encoder?

---

> ### Author Response · Authors · 2024-11-21
> **Response to reviewer rQZZ**
>
> We thank reviewer rQZZ for the thoughtful comments. We now address them individually:
>
> > Dependence on Large EHR Datasets: This approach relies on extensive, high-quality EHR data, which many medical datasets do not include.
>
> EHR is universal in all hospitals (30% of world data volume is produced in the healthcare system [1]). So in practice, EHR data is always available when CT is captured in any healthcare setting realistically. For the lack of such datasets in the research community, we hope our work can spark interest in the community to curate larger scale medical datasets with EHRs to enhance the research along this line. We modified the Introduction section to reflect this.
>
>
> > Limited Modality Scope: Tested only on CT images; broader modality testing could validate versatility across imaging types.
>
> > Why start from 3D image scans instead of 2D medical images? Is this due to the dataset choice, or has similar work already been done on 2D data?
>
> We agree with this comment. Currently there are few if not zero medical imaging datasets that have patient-level linkage for EHR, medical images and prognosis labels. We used INSPECT given it offers all these and it happens to be a 3D dataset. In addition, 3D medical imaging data has richer biomarkers and thus better suited for opportunistic clinical decision tools, especially for detection of underdiagnosed conditions (unlike 2D imaging, which offers flat, linear views limited to length and width, 3D imaging adds depth, providing a comprehensive representation in three dimensions: length, width, and height (x, y, z coordinates)) [2][3]. We argue that our work is to proposing a method, that can readily expand in both 2D and 3D as well as other types of imaging such as X-ray, ultrasound, echocardiogram, MRI and whole slide imaging (WSI).
>
>
> > Interpretability: The TTE pretraining’s impact on specific pixel-level biomarkers is less clear; additional analysis on feature attribution could help.
>
> We have added the Appendix P to illustrate the feature attribution using GradCAM. We can observe that the TTE version of the model can focus the pathology in a more reasonable region rather than scattered features learned by different baselines.
>
>
> > How does the choice of time segmentation for EHR data affect model results during pretraining? Specifically, my understanding is that the model predicts the probability of a patient experiencing a certain event at intervals like 1, 2, or 3 years post-scan. How does the granularity of these time segments impact the performance of the pretrained encoder?
>
>
> Per the findings from related work [4] (see their Figure 3), different # pieces lead to minimal differences in performance so we chose 8 time pieces for PEANN formulation. In addition, we did an ablation testing on adaptation stage with 8 bins of PEANN head, and the comparison against DeepSurv (essentially 1 time bin) is close and within a margin of error, so we decided to stick with the more publicly available DeepSurv package for our adaptation (which provides risk hazard as one score rather than a piece wise manner).
> For pretraining, each time piece is not from a preset threshold like 1, 2 or 3 years post-scan but a cohort dependent metric, i.e. time_piece = (max_time_in_cohort - min_time_in_cohort) / 8. We added explanations in the Experiment section, Setup subsection.
>
>
> [1] Thomason, Jane. "Big tech, big data and the new world of digital health." Global Health Journal 5.4 (2021): 165-168.
>
> [2] Aali, Asad, et al. "Detecting Underdiagnosed Medical Conditions with Deep Learning-Based Opportunistic CT Imaging." arXiv preprint arXiv:2409.11686 (2024).
>
> [3] Pickhardt, Perry J. "Value-added opportunistic CT screening: state of the art." Radiology 303.2 (2022): 241-254.
>
> [4] Steinberg, E., Fries, J. A., Xu, Y., & Shah, N. (2024). MOTOR: A Time-to-Event Foundation Model For Structured Medical Records. In The Twelfth International Conference on Learning Representations.

---

> ### Author Response · Authors · 2024-11-26
>
> Hi reviewer rQZZ,
>
> We thank you for your thoughtful comments and constructive feedback. We are wondering if our rebuttal addresses your concerns regarding:
>
> - Dataset selection
> - CT modality
> - Interpretability of model
> - Time segment choices
>
> The discussion period is coming to an end and we hope the new manuscript brings more clarity to the comments. If not, what other aspect we can improve on?
>
> Regards,
>
> The authors

---

> > ### Comment · Reviewer_rQZZ · 2024-11-27
> >
> > I have reviewed the authors' rebuttal and appreciate the detailed explanations provided. My mostly concerns have been addressed, and I think this is an solid paper. I will maintain my original score but will increase my confidence score. I look forward to seeing the authors' future work.

---

### Author Response · Authors · 2024-11-21
**General response to all reviewers**

Dear reviewers,

We thank you for the constructive feedback and questions. We appreciate all the reviewers for their thoughtful feedback and constructive criticisms of our manuscript. We are grateful for all reviewers for acknowledging:
- Innovative approach (rQZZ, Wo2C, iczn),
- Presentation and flow of the paper (Wo2C, iczn)
- Comprehensive experiment (rQZZ, iczn)
- Background and discussion strength (iczn)

At the same time, we appreciate the criticism and suggestions for our paper:
- Dependency on EHR data (rQZZ, Wo2C)
- Input modality as only CT (rQZZ)
- Interpretability on feature attribution (rQZZ) and label construction (iczn)
- Time-to-event objective clarity, including time piece segmentation (rQZZ), details of 8,192 pretraining tasks (iczn)
- Lack of pretraining baselines and wider model architectures (Wo2C)
- Lack of private data for external validation (Wo2C)

In summary, we have added respective sections in our new manuscript to address these concerns:
- We have rewritten the Introduction section to address the core contribution claims
- We added Figure 1 to elucidate the state of affairs, where existing methods do not take EHR into account but only local/short-horizon information as supervision, framed as a missing context problem. We further changed the paper title to reduce the confusion brought by reviewers.
- We added Appendix P  for feature attribution
- We added Appendix O and Appendix N to illustrate the examples of pretraining tasks and the aggregated distribution of event times in our cohort
- We added Appendix L to detail the piecewise exponential loss function formulation
- We added Appendix D for baseline comparison with a current CT foundation model Merlin’s performance
- We reorganized the evaluation metrics. Specifically we added Harrell’s C-index as TTE evaluation in Table 3 (which doesn’t evaluate on segmented timepieces but one hazard score in non-piecewise fashion) and moved time-dependent C-statistics into Appendix C, given this metrics up-weighs the long time horizon examples and thus gives unfair evaluation on non-piecewise models (i.e. our CoxPH survival head).

In addition to that, we reorganized the paper and added extra contents to make the manuscript more clear:
- We conducted our classification task with Logistic Regression to align common practices of ‘linear probing’ for foundation model adaptation (therefore expect the Table 2 and 4 to have updated numbers)
- We added Appendix I for full parameter fine-tuning experiment to show the generalizability of our pretraining method compared to per-task fine-tuning
- We added Append H for computation cost of our models
- We added Appendix M for Kaplan-Meier curves to illustrate long term survival probabilities for respective TTE tasks

We have highlighted the changed text to the color of orange for ease of comparison.

---

### Comment · Area_Chair_qZPa · 2024-11-25
**Please engage in discussion**

Dear all,

Many thanks to the reviewers for their constructive reviews and the authors for their detailed responses.

Please use the next ~2 days to discuss any remaining queries as the discussion period is about to close.

Thank you.

Regards,

AC

---

### Meta-Review · Area_Chair_qZPa · 2024-12-20

**Metareview:**

The paper presents a novel framework called “Time-to-Event (TTE) pretraining” for 3D medical imaging models, aiming to improve outcome prediction by leveraging longitudinal supervision from electronic health records (EHRs). Traditional pretraining methods for 3D medical imaging focus on capturing structural features, but they fail to establish links between pixel-level biomarkers and future health outcomes due to the absence of temporal context. To address this, the authors propose a method that incorporates future event information from EHRs into the pretraining process.

The key contribution of the paper is the integration of time-to-event modelling with 3D imaging data. Using a large dataset of 18,945 chest CT scans linked to longitudinal EHR data, the authors constructed 8,192 unique pretraining tasks. Each task predicts not only whether a clinical event will occur but also when it is likely to happen, using distributions derived from EHR records. This approach significantly increases label density, resulting in a richer and more informative pretraining signal. The authors report good performance improvements, achieving a 23.7% increase in AUROC and a 29.4% improvement in Harrell’s C-index across eight benchmark tasks, while maintaining strong performance on standard diagnostic classification tasks. This demonstrates the framework’s capacity to better identify prognostic pixel biomarkers in 3D medical images.

The strengths of the proposed approach are its scalability, use of large-scale multimodal data, and enhanced predictive performance. By incorporating temporal supervision, the authors provide a more comprehensive view of disease progression, enabling models to capture subtle imaging biomarkers that are associated with future health risks. The methodology also handles right-censored data, a common challenge in survival analysis, and utilises large datasets to pretrain models, contributing to generalisability. The use of open datasets and publicly available code promotes transparency and reproducibility in the research community.

However, the study has certain limitations. First, the reliance on large computational resources poses practical challenges for institutions without access to high-memory GPUs, as highlighted by the authors themselves too. Additionally, the pretraining dataset, while substantial, is still smaller than datasets used in other large-scale model pretraining efforts, which may affect generalisation across different populations and imaging modalities. The study is also focused on CT scans, possibly limiting its applicability to other imaging types like MRIs or X-rays; but that is secondary.

In conclusion, the paper offers a decent contribution to the field of medical imaging by introducing a time-to-event pretraining strategy that bridges the gap between imaging data and long-term clinical outcomes.

**Additional Comments On Reviewer Discussion:**

The paper received positive feedback from reviewers, with most leaning toward acceptance after a period of discussion and revisions. While several key concerns were raised initially, the authors provided detailed responses and revisions that ultimately led to an improved manuscript and stronger support from the reviewers.

Reviewer Wo2C raised significant concerns regarding the reliance on large-scale longitudinal electronic health record (EHR) data. They argued that this dependency might limit the applicability of the proposed method since not all institutions have access to such comprehensive datasets. In response, the authors highlighted that EHR data is almost universally available in hospitals and is routinely captured alongside medical imaging. They also emphasised that the goal of their approach is to inspire the community to develop larger datasets that integrate imaging and EHR data. To support this point, the authors revised the Introduction to stress the accessibility of EHR data in healthcare settings. While Reviewer Wo2C acknowledged the authors’ rationale, they maintained some reservations and maintained their score (5).

Reviewer rQZZ also raised questions about the dataset scope, particularly the focus on CT scans, which might limit the generalisability of the method to other imaging modalities like MRIs or X-rays. The authors explained that CT scans offer richer biomarkers than 2D imaging modalities, making them a more suitable starting point for this work. However, they emphasised that their approach could be applied to other modalities in future studies. They incorporated this clarification into the paper and justified the choice of CT as a pragmatic design decision. Reviewer rQZZ found the response convincing and noted that the generalisation argument was adequately addressed.

Reviewers Wo2C and iczn critiqued the paper’s lack of comparisons with self-supervised learning (SSL) baselines. They highlighted that the comparison with Merlin was insufficient, as it was not a true 3D pretraining model. Both reviewers requested a broader comparison with other self-supervised models, especially those pre-trained on 3D medical imaging data. In response, the authors provided a more comprehensive comparison with Merlin, demonstrating that their approach outperformed Merlin in 9 out of 24 tasks and was statistically indistinguishable in 10 others. Additionally, the authors justified their choice of model architectures, which included SwinUNETR, ResNet, and DenseNet, arguing that these models cover a range of architecture types. They also noted that the pretraining method is architecture-agnostic, which was reflected in the manuscript’s revisions. This clarification, particularly regarding the architecture choices, was well-received by Reviewer iczn, who acknowledged the technical breadth and rigour of the comparisons. Consequently, Reviewer iczn raised their score from 6 to 8, indicating strong support for acceptance.

A particularly detailed discussion emerged regarding the definition of the 8,192 pretraining tasks and the potential for label leakage. Reviewer iczn sought clarity on how these pretraining tasks were defined, the kinds of events they represented, and whether any of the downstream evaluation labels were present in the pretraining set. To address this, the authors provided significant additional detail in Appendix O, where they stratified the tasks into frequency-based quintiles and shared illustrative examples, including tasks based on LOINC, RxNorm, and ICD codes. The authors emphasized that no downstream evaluation labels were included in the pretraining task set and explicitly mentioned this clarification in the main text. This addressed Reviewer iczn’s concerns, and they noted that the issue had been resolved to their satisfaction. This clarification contributed to Reviewer iczn’s decision to raise their score to 8.

Another issue from Reviewer Wo2C concerned the use of TTE pretraining and requested a clearer explanation of the methodology. The authors addressed this by adding more detail to Section 4 and expanding Appendix L, where they provided a comprehensive explanation of the piecewise exponential loss function used in TTE modelling. The authors further included a plain-language motivation for the method and clarified that the TTE modelling process treats each task independently while considering death as the only competing risk.

Another issue was the improper display of CT images. Reviewer Wo2C pointed out that some CT images were rotated or windowed incorrectly, which they described as “not a professional way of showing CT images.” The authors redesigned and updated these figures. They also opted to use cube-shaped visual representations instead of 2D projections, making it clear that the figures were meant to illustrate concepts rather than represent actual scan images.

Overall, the paper has improved considerably throughout the rebuttal period and is worth publishing.

---

### Decision · Program_Chairs · 2025-01-22

Accept (Poster)